# Video compression dataset and benchmark of learning-based video-quality metrics

**Anastasia Antsiferova**[1,2][*], **Sergey Lavrushkin**[1,2][*], **Maksim Smirnov**[3][*],
**Alexander Gushchin**[3][*], **Dmitriy Vatolin**[1,2,3][*], **Dmitriy Kulikov**[2,3][*]
ISP RAS Research Center for Trusted Artificial Intelligence[1]
MSU Institute for Artificial Intelligence[2]
Lomonosov Moscow State University[3]
{aantsiferova, sergey.lavrushkin, maxim.smirnov.2025,
alexander.gushchin, dmitriy, dkulikov}@graphics.cs.msu.ru

## Abstract

Video-quality measurement is a critical task in video processing. Nowadays, many implementations of new encoding standards — such as AV1, VVC, and LCEVC — use deep-learning-based decoding algorithms with perceptual metrics that serve as optimization objectives. But investigations of the performance of modern video- and image-quality metrics commonly employ videos compressed using older standards, such as AVC. In this paper, we present a new benchmark for video-quality metrics that evaluates video compression. It is based on a new dataset consisting of about 2,500 streams encoded using different standards, including AVC, HEVC, AV1, VP9, and VVC. Subjective scores were collected using crowdsourced pairwise comparisons. The list of evaluated metrics includes recent ones based on machine learning and neural networks. The results demonstrate that new no-reference metrics exhibit high correlation with subjective quality and approach the capability of top full-reference metrics.

## 1  Introduction

Video constitutes the largest part of the world's Internet traffic, and its volume has increased because of the Covid lockdowns. The network load has also increased, making efficient video compression extremely important. Development and comparison of new video encoders greatly relies on quality measurement, and many new compression standards implement machine-learning- and neural-network-based approaches. But traditional image- and video-quality metrics, such as PSNR and SSIM, emerged long before recent compression standards, and they did not account for neural-network-related artifacts. VMAF [26], a well-known video-quality metric from Netflix, was also trained using only H.264/AVC-compressed videos. Thus, quality measurement for new video-encoding standards is even more vital. The number of new image- and video-quality metrics has increased, and many recent algorithms employ learning-based approaches. Industry leaders have also created their own quality metrics: Apple's Advanced Video Quality Tool (AVQT) [2], Tencent's Deep Learning-Based Video Quality Assessment (DVQA) [1], and the aforementioned VMAF. Only a few of these metrics demonstrate high performance on independent benchmarks, however, and some new ones, including AVQT and DVQA, still await detailed analysis. A concern associated with

---

[*]A. Antsiferova designed methodology and performed results analysis, led the benchmark and the paper preparation, S. Lavrushkin performed results analysis and the paper preparation, M. Smirnov developed the benchmark of no-reference metrics, A. Gushchin developed the benchmark of full-reference metrics, both of them performed the dataset research, results analysis and contributed to the paper preparation, D. Vatolin and D. Kulikov coordinated the benchmark and the dataset creation

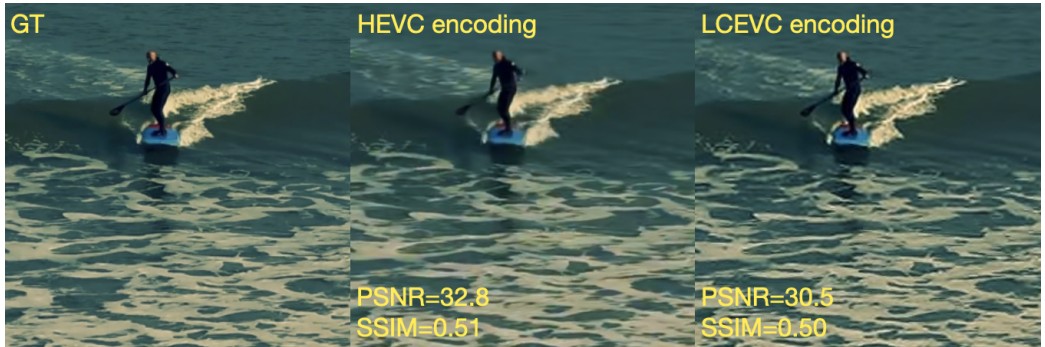

Figure 1: Crops from video sequences encoded using old and new standards relative to ground truth (GT). LCEVC employs super-resolution, which allows restoration of more details and creates new kinds of artifacts.

metric-result reproducibility and verification is the outdated datasets for measuring video-compression quality. Most datasets containing compressed videos and subjective scores only employ H.264/AVC compression. In-lab tests were the source of subjective scores for many such videos. Owing to the complexity and high cost of subjective comparisons, those tests involved a small number of viewers and garnered only a few scores per video.

Quality-metric development seldom takes into account artifacts produced by video encoders that implement contemporary standards. For example, super-resolution in LCEVC and in new neural-network-based encoders yields distortions that traditional metrics are unable to handle. Fig. 1 demonstrates the difference between frame crops of x265-encoded video and lcevc_x265-encoded video: the latter contains more detail despite its lower PSNR and SSIM scores. Existing benchmarks for image- and video-quality metrics do not consider artifacts produced by new compression standards. Our research therefore analyzed metric performance on videos with various compression artifacts.

The goal of our investigation was to evaluate new and state-of-the-art image- and video-quality metrics independently, using a large dataset representing diverse compression artifacts from different video encoders. We thus propose a new dataset of 2,486 compressed videos and subjective scores collected using a crowdsourced comparison with nearly 11,000 participants. We also present a new benchmark[2] based on that dataset, which we divide into open and hidden parts. This paper provides our assessment results for the open part as well as for the whole dataset.

## 2 Related Work

### 2.1 Video-Quality Datasets

Video-quality datasets with subjective scores break down into two types: legacy synthetically distorted (mainly through compression and transmission distortions, capture impairments, processing artifacts, and Gaussian blur), and authentic user-generated content (UGC). The former [43, 11, 42, 7, 32, 27, 37, 19] apply synthetic distortions to the original videos. The latter [14, 39, 36, 16, 44, 50] are gaining popularity, as videos produced today by amateurs often suffer from a wide variety of distortions. Many new video-quality metrics have undergone testing only for UGC videos. The latest studies employ a nearly identical pool of subjective video-quality datasets, summarized in Tab. 1.

### 2.2 Video-Quality Benchmarks

Most comparisons of IQA and VQA have appeared in papers that present new methods and a few benchmarks accept new methods for evaluation. Often, these comparisons either include an evaluation using open video datasets, for which existing metrics may have been tuned, or employ just a few methods. The authors of [39] published a comparison for a wide variety of datasets but evaluated only four methods. In [41] the authors compared no-reference VQA models using three UGC datasets

---

[2]https://videoprocessing.ai/benchmarks/video-quality-metrics.html

| Dataset | Original videos | Average duration (s) | Distorted videos | Distortion | Subjective framework | Subjects | Answers |
|---|---|---|---|---|---|---|---|
| MCL-JCV (2016) [42] | 30 | 5 | 1,560 | Compression | In-lab | 150 | 78K |
| VideoSet (2017) [43] | 220 | 5 | 45,760 | Compression | In-lab | 800 | - |
| UGC-VIDEO (2020) [25] | 50 | > 10 | 550 | Compression | In-lab | 30 | 16.5K |
| CVD-2014 (2014) [36] | 5 | 10-25 | 234 | In-capture | In-lab | 210 | - |
| LIVE-Qualcomm (2016) [14] | 54 | 15 | 208 | In-capture | In-lab | 39 | 8.1K |
| GamingVideoSET (2018) [9] | 24 | 30 | 576 | Compression | In-lab | 25 | - |
| KUGVD (2019) [8] | 6 | 30 | 144 | Compression | In-lab | 17 | - |
| KoNViD-1k (2017) [16] | 1,200 | 8 | 1,200 | In-the-wild | Crowdsource | 642 | 205K |
| LIVE-VQC (2018) [39] | 585 | 10 | 585 | In-the-wild | Crowdsource | 4,776 | 205K |
| YouTube-UGC (2019) [44] | 1,500 | 20 | 1,500 | In-the-wild | Crowdsource | >8,000 | 600K |
| LSVQ (2020) [50] | 39,075 | 5-12 | 39,075 | In-the-wild | Crowdsource | 6,284 | 5M |
| Our dataset: open part (2022) | 36 | 10, 15 | 1,022 | Compression (32 codecs) | Crowdsource | 10,800 | 320K |
| Our dataset: hidden part (2022) | 36 | 10, 15 | 1,464 | Compression (51 codecs) | Crowdsource | 10,800 | 446K |
| Our dataset (2022) | 36 | 10, 15 | 2,486 | Compression (83 codecs) | Crowdsource | 10,800 | 766K |

Table 1: Summary of subjective video-quality datasets and our new dataset.

| Benchmark | Total number of videos | Total number of VQA methods | Total number of subjects | Distortion |
|---|---|---|---|---|
| Z. Sinno and A. Bovik (2018) [39] | 585 | 4 | 4,776 | In-the-wild videos, 80 mobile cameras, 18 resolutions |
| Y. Li *et al.* (2020) [24] | 550 | 15 | 28 | H.264, H.265 compression, QP: 22, 27, 32, 37, 42 |
| UGC-VQA (2021) [41] | 3,108 (LIVE-VQC, YouTube-UGC, KoNViD-1k) | 13 | >13,000 | Compression, transmission |
| Our benchmark (2022) | 2,486 | 26 | 10,800 | Compression (H.264, H.265, AV1, VVC, etc.) |

Table 2: Summary of video-quality-measurement benchmarks and our new benchmark.

and various experiments. They analyzed metrics applied to videos with different content types, resolution and quality subsets, temporal pooling, and computational-complexity-evaluation methods. Compression artifacts, however, played a minor role in that study. The main idea of [24] was to compare full- and no-reference metrics through subjective evaluation of UGC videos transcoded using different compression standards and levels, but this work only tested a few no-reference methods and codecs.

## 3 Benchmark

### 3.1 List of Metrics

This study aimed to evaluate new and state-of-the-art neural-network-based video- and image-quality metrics on a compression-oriented video dataset. We excluded several well-known metrics such as BRISQUE [33] and VIIDEO [34] because of their low correlations in many other studies [50, 22, 24].

#### 3.1.1 No-Reference Video-Quality Metrics

The no-reference video-quality metric **VIDEVAL** (2021) [41] chooses 60 features (related to motion, certain distortions, and aesthetics) from previously developed quality models. It performs well on existing UGC datasets, but it may suffer from overfitting, as users must set many of its parameters.

Most recent quality-assessment papers emphasize deep-learning-based approaches. **MEON** (2017) [29] is a model consisting of two sub-networks: a distortion-identification one and a quality-prediction one. It can also determine the distortion type.

**VSFA** (2019) [20] employs a pretrained ResNet-50 [15] as well as a deep content-aware feature extractor followed by a temporal-pooling layer for temporal memory. It performed poorly in the cross-dataset evaluation, so the authors proposed an enhanced version, **MDTVSFA** (2021) [22]. This enhanced version follows a mixed-dataset training strategy and may have high computational complexity owing to recurrent layers and full-size-image inputs.

**PaQ-2-PiQ** (2020) [51] uses a deep region-based architecture trained on a large subjective image-quality dataset of 40,000 pictures. **KonCept512** (2020) [17] is based on InceptionResNetV2 and was trained on the proposed KonIQ-10k dataset. **SPAQ** (2020) [13] implements three extra modifications of its baseline model: EXIF-data processing (MT-E), image-attribute observation (MT-A), and observation of a scene's high-level semantics (MT-S). The creators of **Linearity** (2020) [21] introduced their own loss function, "norm-in-norm", which converges 10 times faster than the MAE and MSE loss functions. **NIMA** (2018) [40] was trained on the large-scale Aesthetic Visual Analysis (AVA) dataset and predicts a quality-rating distribution.

### 3.1.2 Full-Reference Video-Quality Metrics

**PSNR** and **SSIM** [45] are among the most popular image- and video-quality metrics. We compared variations of SSIM and **MS-SSIM** [46] in our benchmark; the latter is an advanced version of the former calculated over multiple scales using subsampling.

**LPIPS** (2019) [52] is based on AlexNet and VGG. We chose a VGG-based version for testing because it serves as a generalization of "perceptual loss" [18]. **DISTS** (2020) [12] was designed to tolerate texture resampling and to be sensitive to structural differences. It combines structure- and texture-similarity measurements for corresponding image embedments and is based on a pretrained VGG network.

Tencent's **DVQA** (2020) [1] is based on the C3DVQA network [49]. It uses 3D convolutional layers to learn spatiotemporal features and 2D convolutional layers to extract spatial information.

The main feature of **FovVideoVDP** (2021) [31] is consideration of peripheral visual acuity. This method models the human visual system's response to temporal changes across the visual field. It can estimate flickering, juddering, and other temporal distortions, as well as spatiotemporal artifacts such as those appearing at different degrees of peripheral vision.

**ST-GREED** (2021) [30] can quantify reference and distorted videos of different frame rates without temporal preprocessing. It offers two primary features: SGreed and TGreed. The latter quantifies the statistics of temporal bandpass responses to both spatial and temporal distortions. The former obtains spatial bandpass responses using a local filtering scheme. Calculation of the final ST- GREED value employs the support-vector regressor.

Nowadays **VMAF** (2018) [26] is one of the most popular VQA metrics. It computes three base features—the detail-loss metric (DLM) [23], visual-information fidelity (VIF) [38], and temporal information (TI)—and combines them with a support-vector regressor. We also evaluated **AVQT** (2021) [2], developed by Apple, but the company has yet to publish any technical information.

### 3.2 Video Dataset

To analyze the relevance of quality metrics to video compression, we collected a special dataset of videos exhibiting various compression artifacts. For video-compression-quality measurement, the original videos should have a high bitrate or, ideally, be uncompressed to avoid recompression artifacts. We chose from a pool of more than 18,000 high-bitrate open-source videos from www.vimeo.com. Our search included a variety of minor keywords to provide maximum coverage of potential results—for example "a," "the," "of," "in," "be," and "to." We downloaded only videos that were available under CC BY and CC0 licenses and that had a minimum bitrate of 20 Mbps. The average bitrate of the entire collection was 130 Mbps. We converted all videos to a YUV 4:2:0 chroma subsampling. Our choice employed space-time-complexity clustering to obtain a representative complexity distribution. For spatial complexity, we calculated the average size of x264-encoded I-frames normalized to the uncompressed frame size. For temporal complexity, we calculated the average P-frame size divided by the average I-frame size. We divided the whole collection into 36 clusters using the K-means algorithm [28] and, for each cluster, randomly selected up to 10 candidate videos close to the cluster center. From each cluster's candidates we manually chose one video, attempting to include different

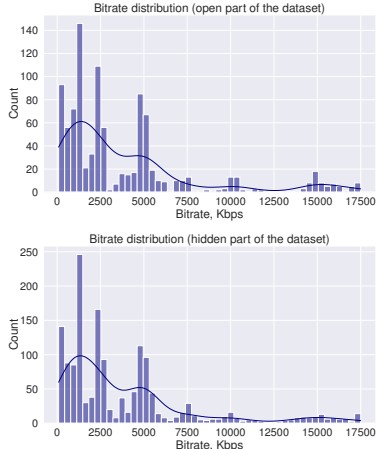

Figure 2: Bitrate distribution of videos in our dataset.

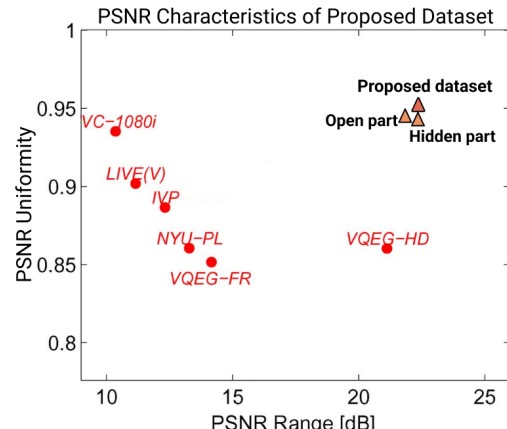

Figure 3: PSNR (range and uniformity) comparison for our dataset versus other video datasets.

genres in the final dataset (sports, gaming, nature, interviews, UGC, etc.). The result was 36 FullHD videos for further compression.

We obtained numerous coding artifacts by compressing videos through several encoders: 11 H.265/HEVC encoders, 5 AV1 encoders, 2 H.264/AVC encoders, and 4 encoders based on other standards. To increase the diversity of coding artifacts, we also used two different presets for many encoders: one that provides a 30 FPS encoding speed and the other that provides a 1 FPS speed and higher quality. The list of settings for each encoder is presented in the supplementary materials. Not all videos underwent compression using all encoders. We compressed each video at three target bitrates — 1,000 kbps, 2,000 kbps, and 4,000 kbps — using a VBR mode (for encoders that support it) or with corresponding QP/CRF values that produce these bitrates. Major streaming-video services recommend at most 4,500–8,000 kbps for FullHD encoding [3, 4, 5]. We avoided higher target bitrates because visible compression artifacts become almost unnoticeable, hindering subjective comparisons. Fig. 2 shows the distribution of video bitrates for our dataset. The distribution differs from the target encoding rates because we used the VBR encoding mode, but it complies with the typical recommendations.

The dataset falls into two parts: open and hidden (40% and 60% of the entire dataset, respectively). We employ hidden part only for testing through our benchmark to ensure a more objective comparison of future applications. This approach may prevent learning-based methods from training on the entire dataset, thereby avoiding overfitting and incorrect results. To divide our dataset, we split the codec list in two; the encoded videos each reside in the part corresponding to their respective codec. We also performed x265-lossless encoding of all compressed streams to simplify further evaluations and avoid issues with nonstandard decoders.

Tab. 1 shows the characteristics of the final parts of the dataset. Links to source videos and additional details about the collection process are in the supplementary materials. We also compared the statistics of PSNR uniformity and range for our dataset using the approach in [47]. As Fig. 3 shows, this dataset provides wide quality and compression-rate ranges.

## 3.3 Subjective-Score Collection

We collected subjective scores for our video dataset through the Subjectify.us crowdsourcing platform. Subjectify.us is a service for pairwise comparisons; it employs a Bradley-Terry model to transform the results of pairwise voting into a score for each video. A more detailed description of the method is at `www.subjectify.us`.

Because the number of pairwise comparisons grows exponentially with the number of source videos, we divided the dataset into five subsets by source videos and performed five comparisons. Each subset contained a group of source videos and their compressed versions. Every comparison produced and

evaluated all possible pairs of compressed videos for one source video. Thus, only videos from the same source were in each pair. The comparison set also included source videos. Participants viewed videos from each pair sequentially in full-screen mode. They were asked to choose the video with the best visual quality or indicate that the two are of the same quality. They also had an option to replay the videos. Each participant had to compare a total of 12 pairs, two of which had an obviously higher-quality option and served as verification questions. All responses from those who failed to correctly answer the verification questions were discarded.

To increase the relevance of the results, we solicited at least 10 responses for each pair. In total, we collected 766,362 valid answers from nearly 11,000 individuals. After applying the Bradley-Terry model to a table of pairwise ranks, we received subjective scores that are consistent within each group of videos compressed from one reference video. A detailed description of the subjective-comparison process, as well as collected statistics, is in the supplementary materials. Tab. 1 summarizes the parameters of our dataset.

## 3.4 Methodology

We used public source code for all metrics without additional pretraining, and we selected the default parameters to avoid overfitting. To get a video's quality score using the IQA method, we compared the given distorted sequence and the reference video frame by frame, then averaged the resulting per-frame quality scores for each video. VQA methods generate a score for the whole distorted sequence and require no additional averaging.

Because the subjective scores are based on pairwise comparisons of videos produced from the same original sequence, they are comparable only within their respective groups. Each group size is three (the number of encoding bitrates) times the number of codecs applied to the reference video. For each reference-video/preset pair (resulting in one distorted-video group), we calculated Spearman and Kendall correlation coefficients (SROCC and KROCC, respectively) between the metrics and subjective scores. We then selected only those values calculated for groups whose number of samples exceed a threshold (15 for SROCC and 6 for KROCC) to provide more-statistically-reliable results. Our next step was to use the Fisher Z-transform [10] (inverse hyperbolic tangent) and average the results, weighted proportionally to group size. The inverse Fisher Z-transform yielded a single correlation for the entire dataset. We provide the link to code example in Sec. 4.

To analyze metric performance in more detail, we added a few mutually nonexclusive categories with videos from the dataset: User-Generated Content, Shaking, Sports, Nature, Gaming / Animation, Low Bitrate (up to 1,000 Kbps), and High Bitrate (above 6,000 Kbps). To assign each video to one of these categories, we conducted a subjective survey of five people from our laboratory.

## 3.5 Results

We examined the results for the open part of the dataset as well as for the whole dataset, including the hidden part. Tab. 3 shows the Spearman and Kendall correlation coefficients for the metrics we analyzed, along with subjective quality scores. For the whole dataset, VMAF and its variations calculated using different chroma-component ratios exhibited the highest correlations. VMAF was originally to be calculated only using the luma component, but here we proved that YUV-VMAF performs better. Also, VMAF NEG (a no-enhancement-gain version [6]) correlated less well with subjective quality than the original version did. MDTVSFA and Linearity had the highest correlations among no-reference methods: about 0.93, nearly matching the top results of full-reference metrics (VMAF at 0.94). For the open dataset, SSIM and PSNR showed the highest correlations in addition to VMAF, followed by a recently-released AVQT by Apple.

We compared metrics using different video subsets: videos with low and high bitrates; videos encoded using HEVC/H.265, AV1, and VVC/H.266 (Tab. 4); and videos with different content types — UGC, Shaking, Sports, Nature, and Gaming/Animation (Tab. 5).

**"High bitrate" and "Low bitrate" encoding**. All metrics showed their lowest correlations for videos encoded at 6,000 Kbps or higher. The reason may be the low confidence of the subjective scores for this category. As we described in Sec. 3.2, viewers apparently have difficulty spotting compression artifacts in videos that employ high-quality encoding. The no-reference MDTVSFA, VSFA, and

| Dataset | All Dataset (Open+Hidden) 2486 videos | | Open Dataset 1022 videos | | Dataset | All Dataset (Open+Hidden) 2486 videos | | Open Dataset 1022 videos | |
|---|---|---|---|---|---|---|---|---|---|
| Metric | SROC | KROC | SROC | KROC | Metric | SROC | KROC | SROC | KROC |
| **No-Reference** | | | | | **Full-Reference** | | | | |
| MEON [29] | 0.507 (0.495, 0.518) | 0.376 (0.367, 0.384) | 0.554 (0.534, 0.574) | 0.350 (0.333, 0.366) | FOV VIDEO [31] | 0.527 (0.521, 0.534) | 0.375 (0.370, 0.380) | 0.565 (0.551, 0.579) | 0.492 (0.477, 0.507) |
| Y-NIQE [35] | 0.599 (0.586, 0.611) | 0.421 (0.411, 0.431) | 0.701 (0.679, 0.721) | 0.557 (0.541, 0.573) | LPIPS [52] | 0.749 (0.742, 0.756) | 0.567 (0.561, 0.573) | 0.787 (0.774, 0.799) | 0.667 (0.655, 0.679) |
| VIDEVAL [41] | 0.729 (0.719, 0.738) | 0.541 (0.532, 0.551) | 0.719 (0.700, 0.737) | 0.558 (0.540, 0.575) | DVQA [1] | 0.763 (0.756, 0.770) | 0.579 (0.572, 0.585) | 0.774 (0.762, 0.786) | 0.683 (0.670, 0.695) |
| KonCept512 [17] | 0.836 (0.831, 0.841) | 0.661 (0.655, 0.666) | 0.861 (0.853, 0.868) | 0.696 (0.688, 0.703) | GREED [30] | 0.764 (0.759, 0.769) | 0.588 (0.582, 0.593) | 0.790 (0.782, 0.797) | 0.644 (0.634, 0.654) |
| NIMA [40] | 0.849 (0.844, 0.854) | 0.675 (0.668, 0.681) | 0.868 (0.860, 0.875) | 0.729 (0.719, 0.738) | Y-VQM [48] | 0.821 (0.815, 0.827) | 0.644 (0.637, 0.651) | 0.881 (0.870, 0.890) | 0.767 (0.756, 0.777) |
| PaQ-2-PiQ [51] | 0.871 (0.866, 0.875) | 0.708 (0.702, 0.714) | 0.901 (0.894, 0.908) | 0.752 (0.743, 0.761) | DISTS [12] | 0.847 (0.842, 0.851) | 0.671 (0.667, 0.676) | 0.873 (0.866, 0.879) | 0.753 (0.744, 0.761) |
| SPAQ MT-A [13] | 0.879 (0.875, 0.884) | 0.715 (0.709, 0.720) | **0.912** (0.905, 0.919) | 0.796 (0.786, 0.805) | AVQT [2] | 0.876 (0.872, 0.879) | 0.720 (0.716, 0.725) | 0.889 (0.882, 0.896) | 0.792 (0.784, 0.800) |
| SPAQ BL [13] | 0.880 (0.875, 0.884) | 0.711 (0.704, 0.717) | **0.912** (0.905, 0.918) | 0.789 (0.780, 0.798) | YUV-PSNR | 0.883 (0.879, 0.887) | 0.728 (0.722, 0.733) | 0.949 (0.946, 0.952) | 0.849 (0.844, 0.854) |
| SPAQ MT-S [13] | 0.882 (0.878, 0.886) | 0.719 (0.713, 0.724) | **0.912** (0.906, 0.918) | 0.787 (0.778, 0.796) | YUV-SSIM | 0.906 (0.902, 0.909) | 0.756 (0.750, 0.761) | 0.948 (0.945, 0.951) | 0.897 (0.872, 0.917) |
| VSFA [20] | **0.905** (0.901, 0.908) | **0.748** (0.743, 0.753) | 0.891 (0.886, 0.897) | 0.758 (0.750, 0.766) | Y-MS-SSIM [46] | 0.909 (0.905, 0.912) | 0.756 (0.751, 0.760) | 0.946 (0.943, 0.949) | 0.841 (0.835, 0.847) |
| Linearity [21] | **0.910** (0.907, 0.913) | **0.759** (0.754, 0.763) | 0.905 (0.899, 0.911) | 0.783 (0.774, 0.791) | Y-VMAF NEG [26] | 0.914 (0.911, 0.917) | 0.765 (0.760, 0.769) | 0.945 (0.942, 0.948) | 0.841 (0.836, 0.846) |
| MDTVSFA [22] | **0.929** (0.927, 0.931) | **0.788** (0.784, 0.792) | **0.930** (0.927, 0.934) | **0.813** (0.806, 0.819) | YUV-VMAF NEG [26] | **0.917** (0.914, 0.920) | **0.769** (0.765, 0.774) | 0.947 (0.944, 0.950) | **0.894** (0.869, 0.915) |
| | | | | | Y-VMAF (v061) [26] | 0.942 (0.940, 0.944) | **0.809** (0.805, 0.813) | 0.945 (0.942, 0.948) | 0.888 (0.861, 0.910) |
| | | | | | YUV-VMAF (v061) [26] | 0.943 (0.941, 0.945) | 0.810 (0.806, 0.814) | 0.948 (0.945, 0.951) | 0.895 (0.870, 0.916) |

Table 3: Results for SROCC and KROCC on the full dataset and on the open part.

Linearity metrics performed better than the full-reference alternatives. The leaders for "Low Bitrate" remain the same as for the whole dataset supplemented by no-reference NIMA.

**HEVC, AV1, and VVC encoding**. Metric correlation for "H.265 encoding" is higher than for other standards. Because H.265 is older and more popular than newer standards, quality-assessment models may have been tuned to it. For the new VVC standard, the leaders differ relative to other encoding standards: the best no-reference metric is Linearity and the best full-reference one is SSIM. This result is unexpected, but SSIM's good performance may owe to its versatility. Also, the sample for this category is small, so further analysis of the best metrics for estimating VVC encoding quality would likely require a larger dataset.

**Leaders by video content category: "UGC", "Shaking", "Sports", "Nature", and "Gaming/Animation"**. VMAF retained its lead in all categories, but its original luma-only (Y-VMAF) version performed better on "UGC" and "Shaking" content; its modified version using chroma components (YUV-VMAF) is the best for other categories. The no-reference MDTVSFA metric leads in "UGC", "Shaking" and "Nature"; Linearity is ahead in "Sports"; and VSFA is best for "Gaming/Animation". PaQ-2-PiQ also achieves to precisely estimate gaming-content quality.

We performed a one-sided Wilcoxon rank-sum test on SROCC, which we computed for most of the methods in Tab. 4 and Tab. 5 using different groups of videos from our dataset to get the average correlation value. A table of results appears in the supplementary materials. Different versions of SPAQ (BL, MT-S, and MT-A) behaved in a statistically equal manner—except in the "Sports" and "Low Bitrate" categories, where SPAQ BL was superior. In addition, VMAF, the top full-reference metric in average SROCC for the full dataset, yielded to MDTVSFA, the leading no-reference metric, only on videos encoded using AV1 and videos with a high bitrate. A rarely used encoding standard for training quality-assessment methods, VVC was difficult for these methods to handle, and videos encoded using it form the only part of our dataset where MDTVSFA fell short of Linearity. Among related metrics, the test revealed that VMAF NEG and VSFA were statistically worse than or equal to VMAF (v061) and MDTVSFA, respectively, depending on the subset. AVQT was superior to most metrics for the "Low Bitrate" and "Shaking" categories, making it valuable when predicting subjective quality.

Tab. 6 shows the computational complexity of the metrics we studied.

Fig. 4 shows the distribution of normalized metric scores for our dataset. Many metrics have a nonuniform "real-life" distribution of values resulting from compression artifacts. For example, the average SSIM is about 0.85, which corresponds with common statistics (an SSIM of 0.5 does not mean average quality, and values below 0.5 seldom appear in real situations).

| Dataset | Low Bitrate (up to 1,000 kbps) 477 videos | | High Bitrate (above 6,000 kbps) 384 videos | | H.265 Encoding 1139 videos | | AV1 Encoding 482 videos | | VVC Encoding 251 videos | |
|---|---|---|---|---|---|---|---|---|---|---|
| Metric | SROC | KROC | SROC | KROC | SROC | KROC | SROC | KROC | SROC | KROC |
| **No-Reference** | | | | | | | | | | |
| MEON [29] | 0.039 | 0.069 | 0.127 | 0.107 | 0.834 | 0.703 | 0.800 | 0.734 | 0.709 | 0.535 |
|  | (0.000, 0.093) | (0.032, 0.106) | (0.097, 0.158) | (0.089, 0.125) | (0.825, 0.842) | (0.661, 0.740) | (0.768, 0.828) | (0.625, 0.815) | (0.662, 0.750) | (0.492, 0.576) |
| Y-NIQE [35] | 0.313 | 0.214 | 0.027 | 0.013 | 0.722 | 0.519 | 0.629 | 0.640 | 0.389 | 0.271 |
|  | (0.263, 0.361) | (0.181, 0.246) | (0.000, 0.063) | (0.000, 0.040) | (0.706, 0.736) | (0.431, 0.597) | (0.574, 0.678) | (0.501, 0.747) | (0.311, 0.462) | (0.218, 0.323) |
| VIDEVAL [41] | 0.615 | 0.415 | 0.290 | 0.209 | 0.804 | 0.661 | 0.754 | 0.625 | 0.635 | 0.490 |
|  | (0.580, 0.647) | (0.387, 0.441) | (0.256, 0.322) | (0.189, 0.229) | (0.791, 0.817) | (0.576, 0.731) | (0.707, 0.794) | (0.596, 0.653) | (0.576, 0.688) | (0.444, 0.533) |
| KonCept512 [17] | 0.891 | 0.719 | 0.204 | 0.164 | 0.904 | **0.876** | 0.819 | **0.990** | 0.849 | 0.693 |
|  | (0.883, 0.899) | (0.708, 0.729) | (0.149, 0.259) | (0.136, 0.192) | (0.898, 0.909) | (0.837, 0.907) | (0.793, 0.842) | (0.972, 0.996) | (0.819, 0.874) | (0.658, 0.725) |
| NIMA [40] | **0.904** | **0.764** | 0.361 | 0.256 | 0.879 | 0.791 | 0.807 | 0.953 | 0.712 | 0.540 |
|  | (0.895, 0.913) | (0.751, 0.776) | (0.315, 0.405) | (0.232, 0.280) | (0.872, 0.886) | (0.748, 0.835) | (0.774, 0.835) | (0.901, 0.978) | (0.648, 0.765) | (0.486, 0.589) |
| PaQ-2-PiQ [51] | 0.880 | 0.689 | 0.402 | 0.291 | 0.911 | 0.861 | 0.870 | 0.978 | 0.888 | **0.749** |
|  | (0.871, 0.888) | (0.675, 0.703) | (0.343, 0.457) | (0.261, 0.321) | (0.906, 0.915) | (0.823, 0.891) | (0.851, 0.886) | (0.949, 0.991) | (0.865, 0.908) | (0.717, 0.777) |
| SPAQ MT-A [13] | 0.842 | 0.689 | 0.393 | 0.308 | 0.898 | 0.816 | 0.870 | 0.957 | **0.894** | 0.727 |
|  | (0.835, 0.849) | (0.677, 0.702) | (0.356, 0.430) | (0.286, 0.331) | (0.892, 0.904) | (0.777, 0.849) | (0.853, 0.886) | (0.909, 0.980) | (0.869, 0.915) | (0.689, 0.760) |
| SPAQ BL [13] | 0.844 | 0.672 | 0.401 | 0.332 | 0.901 | 0.820 | 0.875 | 0.888 | 0.887 | 0.729 |
|  | (0.835, 0.852) | (0.659, 0.685) | (0.358, 0.442) | (0.307, 0.356) | (0.894, 0.907) | (0.781, 0.852) | (0.858, 0.890) | (0.812, 0.935) | (0.862, 0.908) | (0.694, 0.760) |
| SPAQ MT-S [13] | 0.810 | 0.648 | 0.417 | **0.344** | 0.891 | 0.808 | 0.882 | 0.959 | **0.908** | **0.764** |
|  | (0.804, 0.816) | (0.640, 0.656) | (0.382, 0.450) | (0.319, 0.368) | (0.883, 0.897) | (0.767, 0.842) | (0.867, 0.896) | (0.914, 0.981) | (0.886, 0.926) | (0.731, 0.793) |
| VSFA [20] | 0.894 | **0.757** | **0.517** | 0.370 | 0.927 | 0.790 | **0.914** | 0.989 | 0.846 | 0.694 |
|  | (0.890, 0.898) | (0.748, 0.764) | (0.467, 0.565) | (0.339, 0.401) | (0.922, 0.932) | (0.782, 0.799) | (0.900, 0.927) | (0.973, 0.996) | (0.818, 0.870) | (0.662, 0.723) |
| Linearity [21] | **0.900** | 0.731 | **0.470** | 0.336 | 0.932 | **0.902** | 0.906 | 0.993 | 0.919 | 0.791 |
|  | (0.894, 0.906) | (0.721, 0.740) | (0.424, 0.514) | (0.311, 0.361) | (0.928, 0.936) | (0.870, 0.926) | (0.892, 0.918) | (0.981, 0.997) | (0.903, 0.933) | (0.768, 0.812) |
| MDTVSFA [22] | 0.943 | 0.818 | 0.560 | 0.363 | 0.945 | 0.871 | 0.932 | 0.997 | 0.882 | 0.746 |
|  | (0.940, 0.946) | (0.811, 0.824) | (0.511, 0.606) | (0.331, 0.394) | (0.941, 0.948) | (0.843, 0.895) | (0.919, 0.943) | (0.991, 0.999) | (0.859, 0.902) | (0.718, 0.772) |
| **Full-Reference** | | | | | | | | | | |
| FOV VIDEO [31] | 0.526 | 0.372 | 0.158 | 0.116 | 0.558 | 0.403 | 0.381 | 0.539 | 0.281 | 0.211 |
|  | (0.512, 0.539) | (0.361, 0.384) | (0.135, 0.182) | (0.100, 0.133) | (0.544, 0.571) | (0.393, 0.413) | (0.349, 0.414) | (0.377, 0.670) | (0.243, 0.319) | (0.185, 0.238) |
| LPIPS [52] | 0.774 | 0.577 | 0.270 | 0.179 | 0.814 | 0.815 | 0.532 | 0.477 | 0.464 | 0.356 |
|  | (0.761, 0.786) | (0.565, 0.589) | (0.246, 0.293) | (0.160, 0.198) | (0.803, 0.824) | (0.757, 0.860) | (0.499, 0.563) | (0.452, 0.502) | (0.422, 0.504) | (0.325, 0.387) |
| DVQA [1] | 0.781 | 0.584 | 0.103 | 0.100 | 0.786 | 0.828 | 0.458 | 0.466 | 0.503 | 0.366 |
|  | (0.766, 0.794) | (0.572, 0.596) | (0.075, 0.130) | (0.088, 0.113) | (0.775, 0.797) | (0.767, 0.874) | (0.434, 0.482) | (0.435, 0.495) | (0.446, 0.555) | (0.327, 0.403) |
| GREED [30] | 0.823 | 0.642 | 0.210 | 0.145 | 0.805 | 0.808 | 0.593 | 0.682 | 0.593 | 0.448 |
|  | (0.811, 0.834) | (0.628, 0.656) | (0.189, 0.231) | (0.128, 0.162) | (0.796, 0.815) | (0.748, 0.854) | (0.562, 0.621) | (0.557, 0.777) | (0.554, 0.630) | (0.417, 0.478) |
| Y-VQM [48] | 0.752 | 0.586 | 0.265 | 0.149 | 0.842 | 0.833 | 0.787 | 0.589 | 0.843 | 0.700 |
|  | (0.721, 0.779) | (0.559, 0.611) | (0.213, 0.315) | (0.117, 0.181) | (0.833, 0.851) | (0.780, 0.873) | (0.754, 0.816) | (0.565, 0.613) | (0.804, 0.874) | (0.656, 0.740) |
| DISTS [12] | 0.901 | 0.731 | **0.417** | **0.245** | 0.866 | 0.873 | 0.711 | 0.731 | 0.626 | 0.460 |
|  | (0.896, 0.906) | (0.723, 0.739) | (0.397, 0.436) | (0.229, 0.260) | (0.860, 0.873) | (0.828, 0.908) | (0.695, 0.726) | (0.621, 0.812) | (0.601, 0.650) | (0.440, 0.478) |
| AVQT [2] | 0.923 | 0.784 | 0.176 | 0.075 | 0.894 | 0.872 | 0.857 | 0.926 | 0.842 | 0.698 |
|  | (0.918, 0.927) | (0.777, 0.791) | (0.129, 0.222) | (0.042, 0.107) | (0.889, 0.899) | (0.831, 0.903) | (0.833, 0.877) | (0.858, 0.962) | (0.812, 0.867) | (0.665, 0.728) |
| YUV-PSNR | 0.907 | **0.869** | 0.239 | 0.119 | 0.893 | 0.911 | 0.813 | 0.641 | **0.900** | **0.773** |
|  | (0.901, 0.912) | (0.814, 0.908) | (0.184, 0.293) | (0.085, 0.152) | (0.886, 0.898) | (0.874, 0.937) | (0.786, 0.837) | (0.616, 0.664) | (0.878, 0.919) | (0.743, 0.800) |
| YUV-SSIM | 0.937 | 0.820 | **0.302** | 0.177 | 0.915 | 0.921 | **0.869** | 0.874 | 0.912 | 0.806 |
|  | (0.935, 0.940) | (0.813, 0.826) | (0.256, 0.347) | (0.144, 0.209) | (0.910, 0.920) | (0.888, 0.944) | (0.848, 0.887) | (0.788, 0.926) | (0.891, 0.929) | (0.776, 0.832) |
| Y-MS-SSIM [46] | 0.952 | 0.897 | 0.259 | 0.134 | 0.910 | 0.901 | 0.860 | 0.819 | 0.905 | 0.778 |
|  | (0.950, 0.955) | (0.854, 0.928) | (0.211, 0.306) | (0.102, 0.166) | (0.905, 0.914) | (0.864, 0.928) | (0.839, 0.879) | (0.741, 0.876) | (0.882, 0.923) | (0.747, 0.805) |
| Y-VMAF NEG [26] | **0.946** | 0.823 | 0.268 | 0.163 | 0.910 | 0.902 | 0.863 | **0.957** | 0.880 | 0.731 |
|  | (0.943, 0.948) | (0.818, 0.827) | (0.215, 0.320) | (0.128, 0.197) | (0.905, 0.915) | (0.865, 0.928) | (0.842, 0.881) | (0.909, 0.980) | (0.855, 0.900) | (0.700, 0.759) |
| YUV-VMAF NEG [26] | 0.945 | 0.836 | 0.245 | 0.146 | 0.920 | 0.925 | 0.861 | 0.884 | 0.896 | 0.766 |
|  | (0.942, 0.947) | (0.831, 0.841) | (0.196, 0.293) | (0.113, 0.179) | (0.915, 0.925) | (0.894, 0.947) | (0.840, 0.880) | (0.806, 0.933) | (0.873, 0.915) | (0.736, 0.793) |
| Y-VMAF (v061) [26] | 0.932 | 0.803 | 0.453 | 0.366 | 0.940 | 0.922 | 0.905 | 0.997 | 0.874 | 0.732 |
|  | (0.928, 0.936) | (0.798, 0.809) | (0.405, 0.499) | (0.337, 0.394) | (0.937, 0.944) | (0.893, 0.943) | (0.890, 0.919) | (0.992, 0.999) | (0.849, 0.895) | (0.701, 0.760) |
| YUV-VMAF (v061) [26] | 0.952 | 0.846 | 0.274 | 0.216 | 0.946 | 0.939 | **0.910** | 0.996 | 0.897 | 0.767 |
|  | (0.950, 0.954) | (0.841, 0.851) | (0.225, 0.322) | (0.186, 0.246) | (0.942, 0.949) | (0.914, 0.957) | (0.894, 0.924) | (0.988, 0.999) | (0.874, 0.916) | (0.737, 0.794) |

Table 4: Results for SROCC and KROCC on five subsets of our dataset (by encoding category).

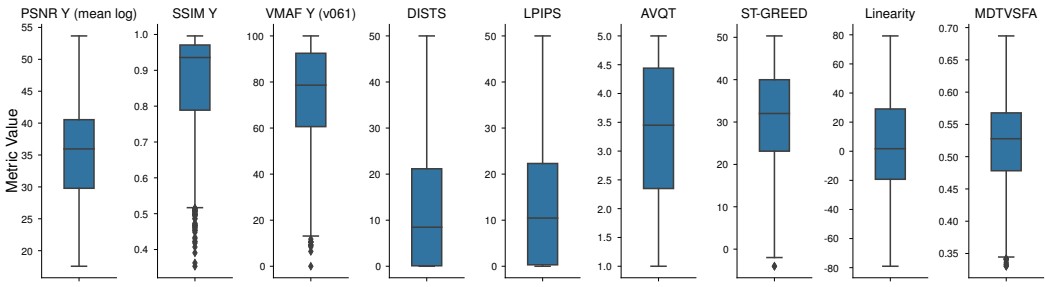

Figure 4: Distribution of metric scores. Each metric appears on a separate axis.

## 4 Conclusion

We created a new diverse dataset containing 2,486 videos compressed by various encoding standards, including AVC, HEVC, AV1, and VVC. We used it to analyze the correlation between new learning-based objective-quality metrics and subjective-quality scores. Our analysis revealed that some new no-reference metrics, such as MDTVSFA, have already caught up with full-reference metrics. At the same time, VMAF showed the highest correlation with subjective scores, making it the best full-reference option for assessing video-compression quality. The open part of the dataset is available publicly [3]. The code, with an example metric launch running on that part of the dataset, is also available [4].

---

[3] https://videoprocessing.ai/datasets/vqa.html

[4] https://github.com/msu-video-group/MSU_VQM_Compression_Benchmark

| Dataset | User-Generated Content 203 videos | | Shaking 594 videos | | Sports 582 videos | | Nature 1216 videos | | Gaming / Animation 204 videos | |
|---|---|---|---|---|---|---|---|---|---|---|
| Metric | SROC | KROC | SROC | KROC | SROC | KROC | SROC | KROC | SROC | KROC |
| **No-Reference** | | | | | | | | | | |
| MEON [29] | 0.072 | 0.102 | 0.221 | 0.176 | 0.645 | 0.490 | 0.524 | 0.405 | 0.707 | 0.531 |
| | (0.021, 0.122) | (0.061, 0.143) | (0.201, 0.241) | (0.162, 0.190) | (0.625, 0.665) | (0.473, 0.507) | (0.508, 0.539) | (0.393, 0.416) | (0.677, 0.734) | (0.503, 0.557) |
| Y-NIQE [35] | 0.232 | 0.164 | 0.473 | 0.332 | 0.672 | 0.485 | 0.621 | 0.443 | 0.810 | 0.623 |
| | (0.148, 0.312) | (0.102, 0.223) | (0.437, 0.507) | (0.305, 0.358) | (0.658, 0.685) | (0.474, 0.496) | (0.604, 0.638) | (0.429, 0.457) | (0.794, 0.826) | (0.602, 0.644) |
| VIDEVAL [41] | 0.422 | 0.304 | 0.535 | 0.386 | 0.836 | 0.645 | 0.739 | 0.548 | 0.923 | 0.773 |
| | (0.365, 0.475) | (0.260, 0.346) | (0.511, 0.559) | (0.367, 0.406) | (0.827, 0.844) | (0.634, 0.655) | (0.726, 0.751) | (0.536, 0.560) | (0.912, 0.933) | (0.751, 0.792) |
| KonCept512 [17] | 0.789 | 0.629 | 0.833 | 0.651 | 0.843 | 0.669 | 0.809 | 0.642 | 0.755 | 0.565 |
| | (0.771, 0.805) | (0.612, 0.646) | (0.819, 0.847) | (0.634, 0.666) | (0.836, 0.849) | (0.661, 0.676) | (0.802, 0.816) | (0.634, 0.649) | (0.732, 0.777) | (0.543, 0.587) |
| NIMA [40] | 0.826 | 0.674 | **0.849** | **0.677** | 0.837 | 0.656 | 0.792 | 0.609 | 0.814 | 0.637 |
| | (0.809, 0.842) | (0.654, 0.693) | (0.838, 0.859) | (0.664, 0.689) | (0.831, 0.843) | (0.649, 0.664) | (0.783, 0.801) | (0.599, 0.618) | (0.804, 0.823) | (0.628, 0.645) |
| PaQ-2-PiQ [51] | 0.801 | 0.659 | 0.807 | 0.643 | 0.908 | 0.753 | 0.827 | 0.658 | 0.963 | 0.855 |
| | (0.778, 0.822) | (0.637, 0.679) | (0.792, 0.821) | (0.627, 0.659) | (0.903, 0.913) | (0.745, 0.761) | (0.818, 0.835) | (0.648, 0.667) | (0.961, 0.965) | (0.851, 0.860) |
| SPAQ MT-A [13] | 0.786 | 0.624 | 0.797 | 0.607 | 0.848 | 0.678 | 0.857 | 0.690 | 0.942 | 0.801 |
| | (0.733, 0.829) | (0.570, 0.672) | (0.776, 0.816) | (0.585, 0.629) | (0.841, 0.854) | (0.671, 0.684) | (0.848, 0.864) | (0.680, 0.700) | (0.937, 0.947) | (0.793, 0.810) |
| SPAQ BL [13] | 0.764 | 0.608 | 0.797 | 0.603 | 0.869 | 0.699 | 0.865 | 0.693 | 0.924 | 0.773 |
| | (0.709, 0.810) | (0.553, 0.657) | (0.773, 0.818) | (0.578, 0.627) | (0.864, 0.875) | (0.693, 0.706) | (0.856, 0.872) | (0.683, 0.704) | (0.921, 0.926) | (0.769, 0.778) |
| SPAQ MT-S [13] | 0.780 | 0.619 | 0.756 | 0.568 | 0.890 | 0.727 | 0.863 | 0.693 | 0.954 | 0.830 |
| | (0.739, 0.815) | (0.575, 0.661) | (0.735, 0.776) | (0.546, 0.589) | (0.884, 0.895) | (0.720, 0.734) | (0.855, 0.871) | (0.683, 0.703) | (0.949, 0.958) | (0.819, 0.839) |
| VSFA [20] | 0.852 | 0.690 | 0.830 | 0.654 | 0.911 | 0.758 | 0.873 | 0.708 | **0.975** | **0.881** |
| | (0.843, 0.861) | (0.678, 0.702) | (0.818, 0.841) | (0.640, 0.667) | (0.905, 0.916) | (0.750, 0.765) | (0.867, 0.878) | (0.701, 0.716) | (0.972, 0.978) | (0.873, 0.889) |
| Linearity [21] | 0.893 | 0.753 | 0.864 | 0.688 | **0.931** | **0.787** | 0.904 | 0.754 | 0.964 | 0.856 |
| | (0.882, 0.903) | (0.738, 0.767) | (0.852, 0.874) | (0.674, 0.702) | (0.927, 0.936) | (0.780, 0.795) | (0.899, 0.909) | (0.747, 0.761) | (0.961, 0.967) | (0.849, 0.862) |
| MDTVSFA [22] | **0.912** | **0.776** | **0.897** | **0.742** | 0.924 | 0.780 | **0.917** | **0.772** | 0.971 | 0.866 |
| | (0.904, 0.918) | (0.763, 0.788) | (0.888, 0.905) | (0.729, 0.754) | (0.920, 0.928) | (0.773, 0.787) | (0.913, 0.921) | (0.766, 0.778) | (0.968, 0.973) | (0.860, 0.872) |
| **Full-Reference** | | | | | | | | | | |
| FOV VIDEO [31] | 0.625 | 0.464 | 0.540 | 0.387 | 0.560 | 0.401 | 0.516 | 0.373 | 0.566 | 0.396 |
| | (0.596, 0.651) | (0.441, 0.486) | (0.528, 0.553) | (0.378, 0.397) | (0.546, 0.574) | (0.389, 0.412) | (0.506, 0.527) | (0.365, 0.380) | (0.550, 0.581) | (0.384, 0.408) |
| LPIPS [52] | 0.724 | 0.587 | 0.639 | 0.473 | 0.854 | 0.674 | 0.746 | 0.565 | 0.785 | 0.597 |
| | (0.692, 0.753) | (0.565, 0.609) | (0.617, 0.660) | (0.455, 0.490) | (0.846, 0.861) | (0.665, 0.683) | (0.734, 0.758) | (0.554, 0.576) | (0.772, 0.798) | (0.586, 0.608) |
| DVQA [1] | 0.847 | 0.689 | 0.708 | 0.528 | 0.841 | 0.653 | 0.783 | 0.600 | 0.883 | 0.709 |
| | (0.824, 0.867) | (0.664, 0.711) | (0.687, 0.727) | (0.510, 0.545) | (0.832, 0.849) | (0.642, 0.663) | (0.772, 0.792) | (0.590, 0.610) | (0.872, 0.894) | (0.694, 0.723) |
| GREED [30] | 0.719 | 0.572 | 0.726 | 0.551 | 0.805 | 0.640 | 0.748 | 0.580 | 0.828 | 0.643 |
| | (0.685, 0.749) | (0.543, 0.600) | (0.712, 0.740) | (0.538, 0.564) | (0.800, 0.811) | (0.634, 0.645) | (0.739, 0.755) | (0.573, 0.588) | (0.812, 0.842) | (0.624, 0.662) |
| Y-VQM [48] | 0.809 | 0.656 | 0.810 | 0.634 | 0.867 | 0.689 | 0.848 | 0.677 | 0.930 | 0.779 |
| | (0.787, 0.830) | (0.637, 0.675) | (0.798, 0.822) | (0.620, 0.647) | (0.857, 0.877) | (0.675, 0.702) | (0.841, 0.855) | (0.669, 0.685) | (0.922, 0.936) | (0.766, 0.791) |
| DISTS [12] | 0.854 | 0.694 | 0.794 | 0.613 | 0.893 | 0.726 | 0.834 | 0.657 | 0.896 | 0.727 |
| | (0.831, 0.874) | (0.668, 0.718) | (0.780, 0.807) | (0.599, 0.627) | (0.888, 0.898) | (0.719, 0.733) | (0.826, 0.841) | (0.649, 0.665) | (0.888, 0.902) | (0.717, 0.737) |
| AVQT [2] | **0.919** | **0.791** | 0.849 | 0.684 | 0.902 | 0.757 | 0.877 | 0.719 | 0.913 | 0.772 |
| | (0.908, 0.928) | (0.774, 0.807) | (0.838, 0.860) | (0.669, 0.698) | (0.896, 0.908) | (0.748, 0.765) | (0.871, 0.883) | (0.712, 0.727) | (0.910, 0.915) | (0.768, 0.776) |
| YUV-PSNR | 0.770 | 0.638 | 0.810 | 0.645 | 0.933 | 0.793 | 0.868 | 0.710 | 0.944 | 0.807 |
| | (0.740, 0.797) | (0.616, 0.658) | (0.797, 0.822) | (0.632, 0.658) | (0.928, 0.937) | (0.785, 0.801) | (0.860, 0.876) | (0.701, 0.719) | (0.938, 0.950) | (0.795, 0.818) |
| YUV-SSIM | 0.779 | 0.642 | 0.811 | 0.648 | **0.952** | **0.828** | 0.900 | 0.750 | 0.958 | 0.837 |
| | (0.750, 0.805) | (0.618, 0.665) | (0.797, 0.824) | (0.633, 0.663) | (0.948, 0.957) | (0.818, 0.837) | (0.893, 0.907) | (0.740, 0.759) | (0.951, 0.964) | (0.823, 0.849) |
| Y-MS-SSIM [46] | 0.895 | 0.746 | 0.851 | 0.680 | 0.942 | 0.808 | 0.901 | 0.746 | 0.955 | 0.832 |
| | (0.882, 0.907) | (0.729, 0.762) | (0.841, 0.862) | (0.666, 0.693) | (0.938, 0.947) | (0.800, 0.817) | (0.895, 0.907) | (0.738, 0.754) | (0.950, 0.960) | (0.821, 0.841) |
| Y-VMAF NEG [26] | 0.916 | 0.779 | **0.861** | 0.688 | 0.945 | 0.810 | 0.909 | 0.757 | 0.960 | 0.841 |
| | (0.907, 0.925) | (0.765, 0.791) | (0.851, 0.870) | (0.675, 0.700) | (0.940, 0.949) | (0.802, 0.818) | (0.903, 0.914) | (0.749, 0.765) | (0.956, 0.964) | (0.832, 0.849) |
| YUV-VMAF NEG [26] | 0.916 | 0.773 | **0.861** | **0.691** | 0.947 | 0.815 | **0.913** | **0.763** | 0.968 | **0.860** |
| | (0.907, 0.924) | (0.761, 0.785) | (0.851, 0.870) | (0.678, 0.703) | (0.942, 0.951) | (0.807, 0.823) | (0.908, 0.918) | (0.756, 0.771) | (0.965, 0.970) | (0.854, 0.866) |
| Y-VMAF (v061) [26] | **0.946** | **0.835** | **0.891** | **0.730** | 0.959 | 0.836 | 0.942 | 0.810 | 0.967 | **0.860** |
| | (0.940, 0.952) | (0.822, 0.846) | (0.882, 0.900) | (0.717, 0.743) | (0.955, 0.962) | (0.828, 0.843) | (0.939, 0.946) | (0.803, 0.816) | (0.964, 0.969) | (0.854, 0.866) |
| YUV-VMAF (v061) [26] | 0.942 | 0.820 | 0.879 | 0.716 | **0.961** | **0.843** | **0.944** | **0.813** | **0.972** | **0.872** |
| | (0.935, 0.948) | (0.807, 0.832) | (0.870, 0.888) | (0.703, 0.729) | (0.958, 0.964) | (0.836, 0.850) | (0.941, 0.947) | (0.806, 0.819) | (0.971, 0.974) | (0.868, 0.876) |

Table 5: Results for SROCC and KROCC on five subsets of our dataset (by content type).

| No-Reference Metric | VIDEVAL[1] (CPU) | MEON[1] (CPU) | Linearity[1] | KonCept512[1] | SPAQ MT-S[1] | SPAQ BL[1] | SPAQ MT-A[1] | NIMA[1] | MDTVSFA[1] | VSF[1]A | PaQ-2-PiQ[1] | NIQE[1] |
|---|---|---|---|---|---|---|---|---|---|---|---|---|
| Computation Complexity (FPS) | 0.62 | 2.27 | 3.41 | 4.18 | 6.48 | 6.49 | 6.66 | 7.24 | 9.12 | 9.26 | 11.10 | 80.00 |

| Full-Reference Metric | LPIPS[1] | DVQA[1] | GREED[1] | DISTS[1] | FOV VIDEO[1] | AVQT (CPU)[2] | VMAF[1] | MS-SSIM[1] | SSIM[1] | VQM[1] | PSNR[1] |
|---|---|---|---|---|---|---|---|---|---|---|---|
| Computation Complexity (FPS) | 3.20 | 5.75 | 7.10 | 8.65 | 37.57 | 37.66 | 52.62 | 99.36 | 160.58 | 280.00 | 371.96 |

Table 6: FPS evaluation for videos from the dataset. The metric testing used a configuration with two Intel Xeon Silver 4216 processors running Ubuntu 20.04 at 2.10 GHz with a Titan RTX GPU, and another configuration with an Intel Core i9 processor running at 2.3 GHz with 16 GB of RAM and AMD Radeon Pro 5500M 4 GB graphics card.

Our proposed dataset will be useful for researchers and developers of image- and video-quality metrics that evaluate video-compression artifacts. It can serve in training models that assess video-compression quality to achieve more-precise results and higher correlation with subjective scores. Our benchmark will remain an unbiased test of compression quality for new image- and video-quality metrics.

We are accepting new methods for evaluation using our benchmark[5]. During the few months since its publication, we have already received several submissions, as well as good reviews and requests for further development. Our plan is to further increase the number of original videos and add new encoders. Because the subjective tests are expensive, we estimate our current dataset cost about $15,000. We are open to collaboration and sponsorship to improve the dataset more quickly and to provide more-reliable and more-valuable results.

---

[5] https://videoprocessing.ai/benchmarks/video-quality-metrics.html

## 4.1 Limitations

We did not retrain the tested metrics on the open part of our dataset. We used already trained models without tuning their parameters. This approach allowed us to prevent metrics from overfitting on our dataset. Nevertheless, some methods are not fitted for data that was absent from the training set (for instance, compression artifacts) or simply underwent training on small datasets. As a result, these metrics may show weak performance on our dataset. Future work will therefore include metric retraining on open part of the dataset and assessment of their quality on the hidden part.

## 4.2 Acknowledgments

The work received support through a grant for research centers in the field of artificial intelligence (agreement identifier 000000D730321P5Q0002, dated November 2, 2021, no. 70-2021-00142 with the Ivannikov Institute for System Programming of the Russian Academy of Sciences).

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
