# Supplementary materials: Video compression dataset and benchmark of learning-based video-quality metrics

**Anastasia Antsiferova**[1,2], **Sergey Lavrushkin**[1,2], **Maksim Smirnov**[3],
**Alexander Gushchin**[3], **Dmitriy Vatolin**[1,2,3], **Dmitriy Kulikov**[2,3]
ISP RAS Research Center for Trusted Artificial Intelligence[1]
MSU Institute for Artificial Intelligence[2]
Lomonosov Moscow State University[3]
{aantsiferova, sergey.lavrushkin, maxim.smirnov.2025,
alexander.gushchin, dmitriy, dkulikov}@graphics.cs.msu.ru

## 1   Appendix

### 1.1   Metric Calculations

Below we describe the steps for calculating metrics. To avoid overfitting on our dataset, we used already fitted image- and video-quality-assessment models with public source code. We left the default parameters of all metrics unchanged.

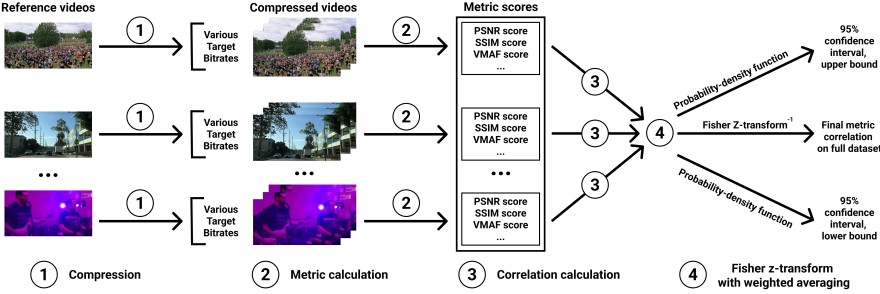

Figure 1: Metric-calculation process.

We tested some metrics (VMAF, PSNR, SSIM, MS-SSIM, VQM, and NIQE) on each color component (Y, U, and V) in addition to averaging the components using different weights. For example, one possibility is SSIM calculated solely on the Y component with a 6:1:1 weighted average.

Below are the steps for calculating different versions of such metrics.

### 1.2   IQA-Method Calculation

We used mean temporal pooling as a way to aggregate scores from multiple frames. Previous research showed no significant difference between pooling methods, and our tests confirmed that finding.

Therefore, to get a quality score for a whole video using an IQA method, we compared a given distorted sequence frame by frame with the corresponding reference video and then averaged the scores. We intend to include more data on this research in future publications.

To perform lossless conversion between file formats, we used the following commands:

36th Conference on Neural Information Processing Systems (NeurIPS 2022) Track on Datasets and Benchmarks.

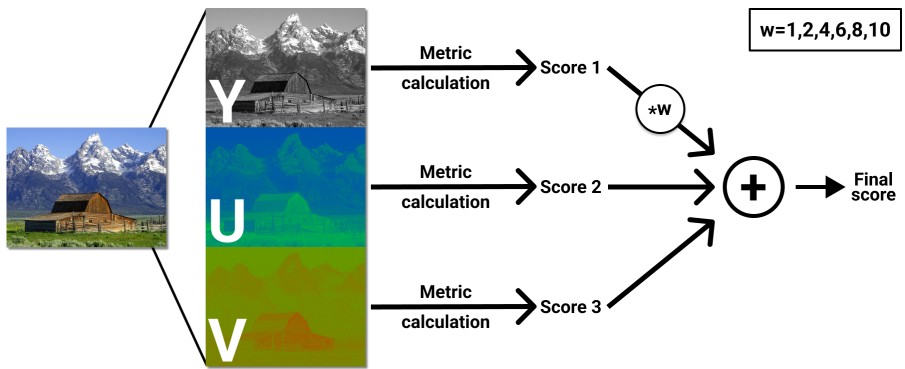

Figure 2: Metric calculation using different weights.

1. .yuv –> .mp4

```
ffmpeg −f rawvideo −vcodec rawvideo −s {width}x{height}
−r {FPS} −pix_fmt yuv420p −i {video name}.yuv −c:v libx265
−x265−params "lossless=1:qp=0" −t {hours:minutes:seconds}
−vsync 0 {video name}.mp4
```

2. .mp4 –> .png

```
ffmpeg −i {sequence name}.mp4 {images dir}/image_%05d.png
```

## 1.3 Metric-Speed Measurement

We also measured the metric-speed performance, expressed in FPS (the execution time of a full model divided by the number of sequence frames).

- The calculation used the following:
    - Five reference videos compressed using the x264 codec (three target bitrates).
    - Three metric calculations for each distorted video.
    - In total, 15 compressed videos and 45 total measurements.
- Output: maximum FPS among three calculations for the given video.
- Calculations employed the following hardware:
    - Nvidia Titan RTX GPU
    - 64-CPU cluster based on Intel Xeon Silver 4216 processor @ 2.10GHz

## 1.4 Metric Correlation for Different Categories

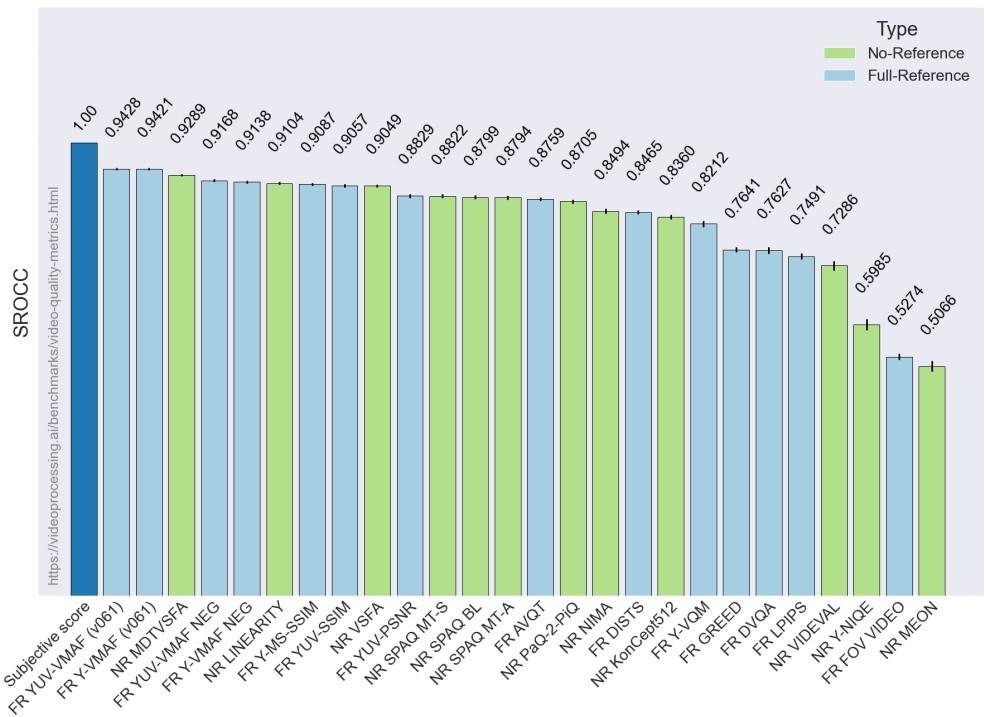

Figure 3: SROCC values on full dataset.

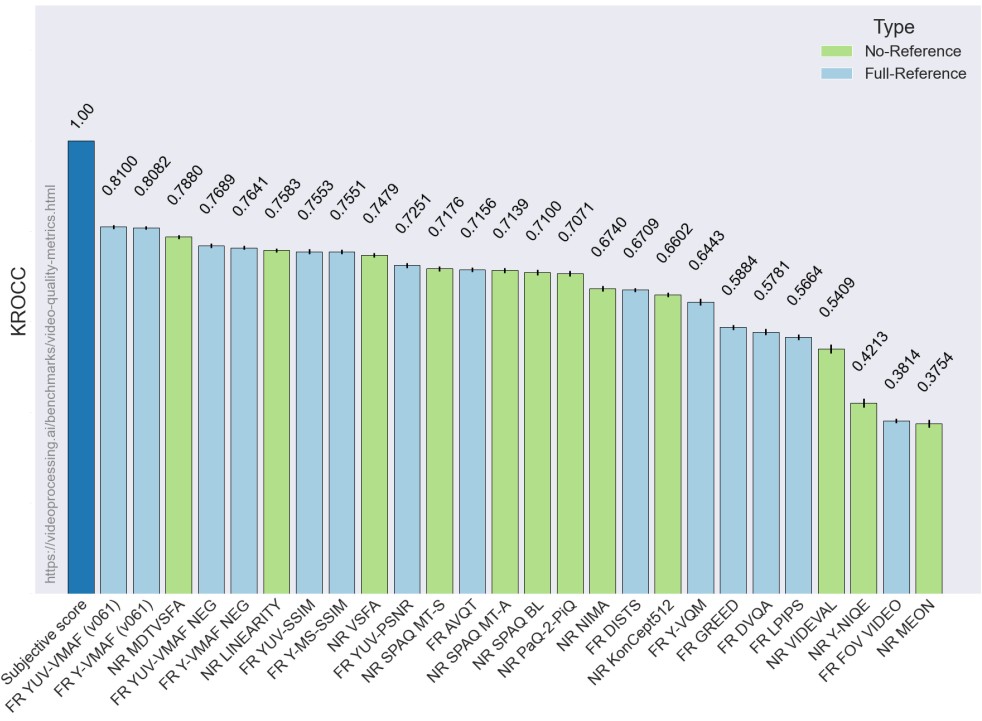

Figure 4: KROCC values on full dataset.

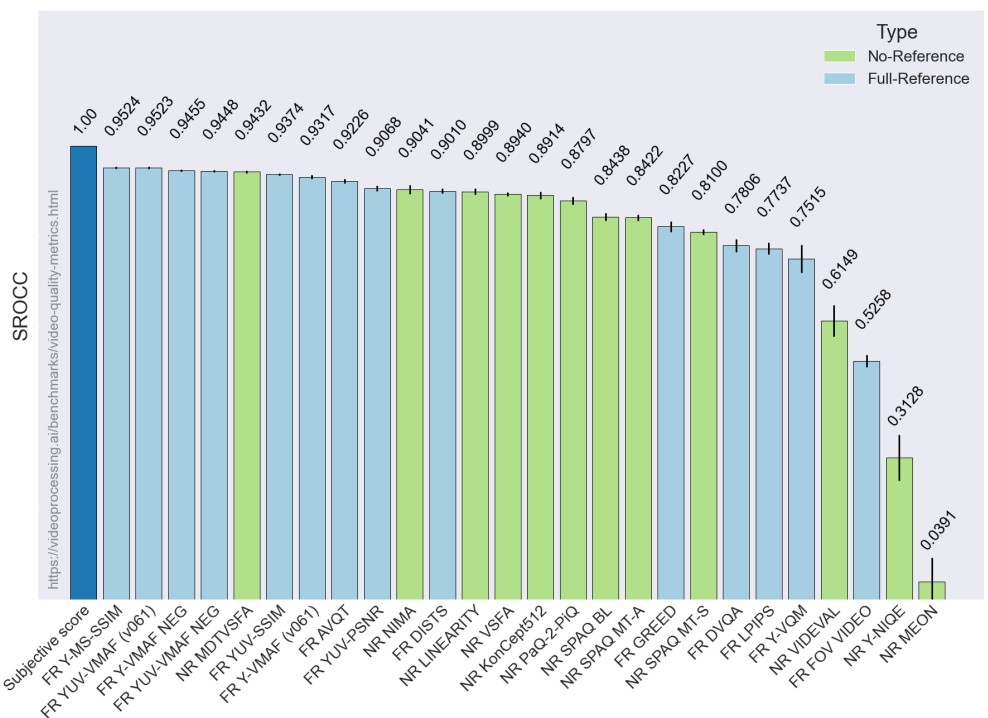

Figure 5: SROCC values for "Low Bitrate" category.

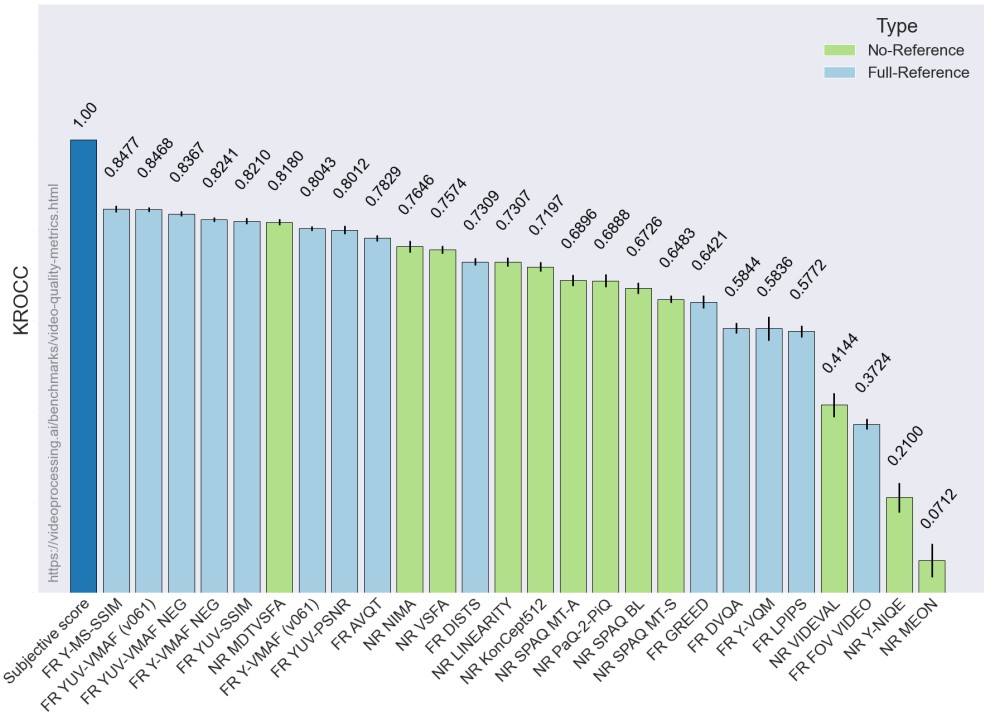

Figure 6: KROCC values for "Low Bitrate" category.

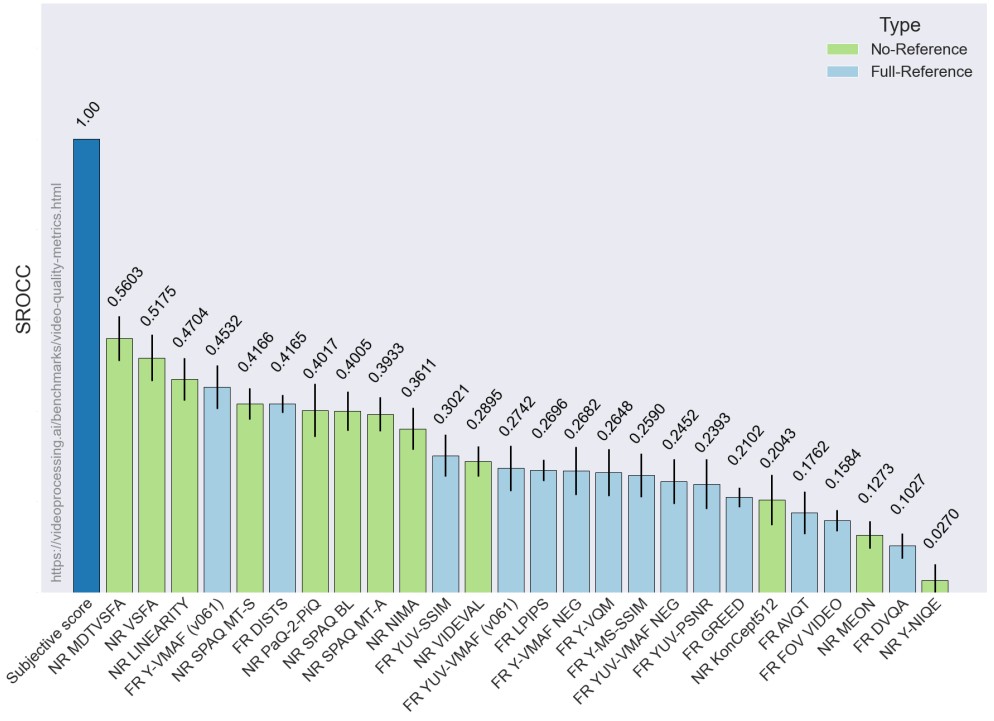

Figure 7: SROCC values for "High Bitrate" category.

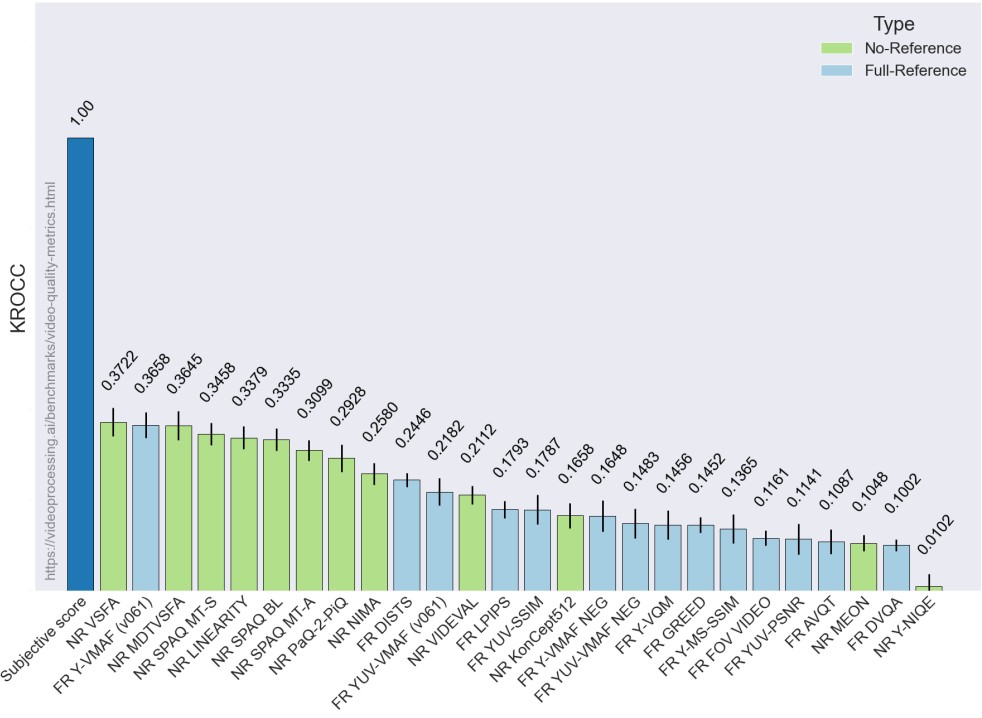

Figure 8: KROCC values for "High Bitrate" category.

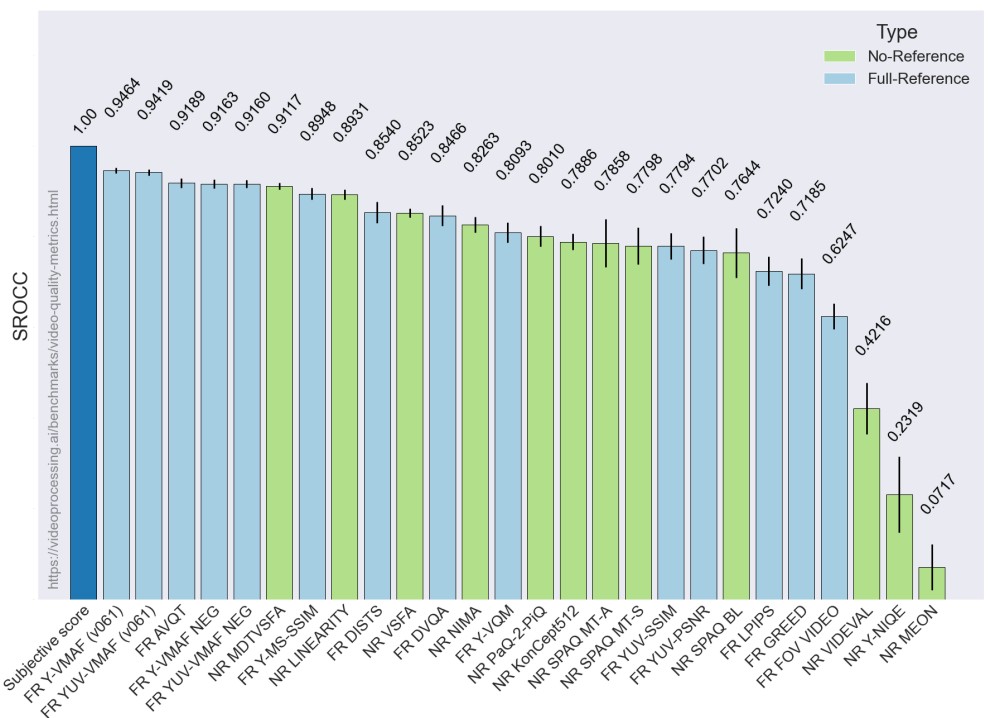

Figure 9: SROCC values for "User-Generated Content" category.

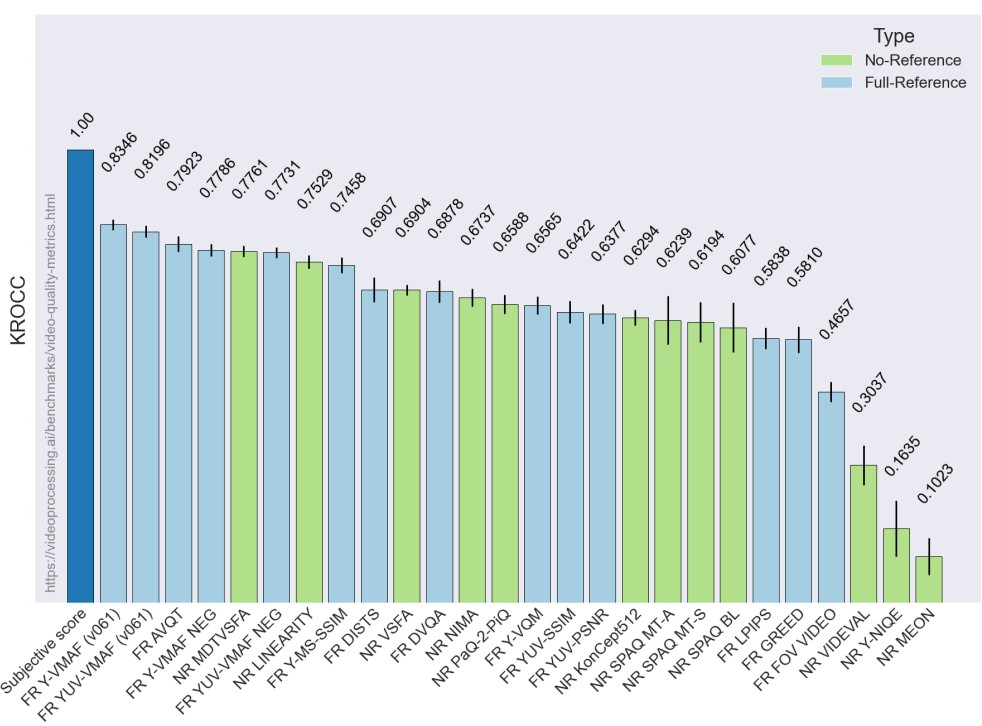

Figure 10: KROCC values for "User-Generated Content" category.

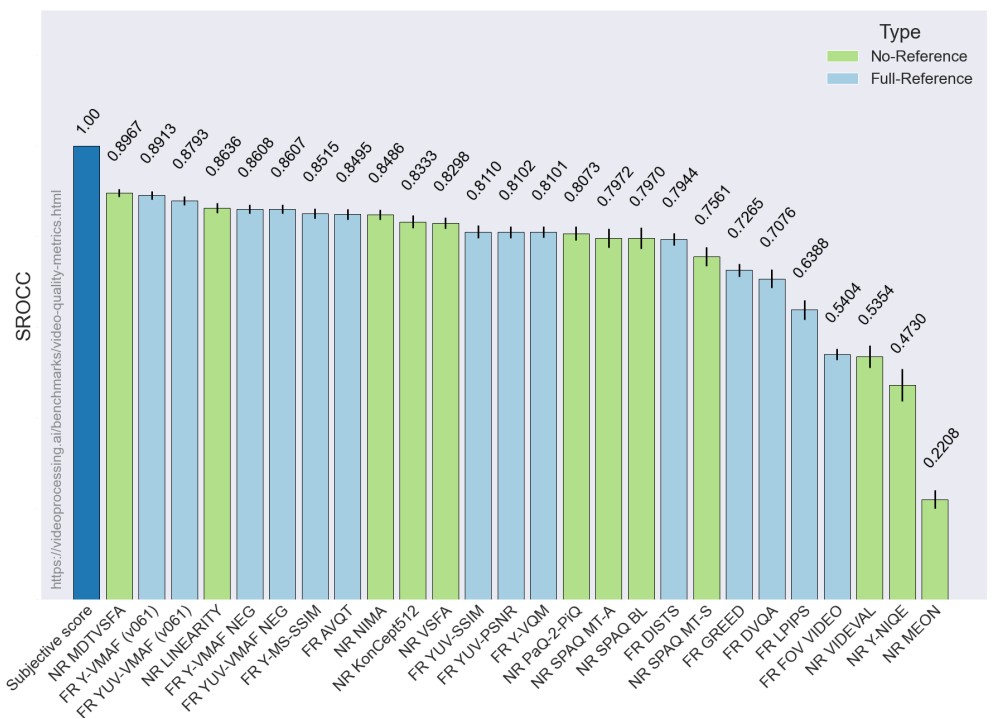

Figure 11: SROCC values for "Shaking" category.

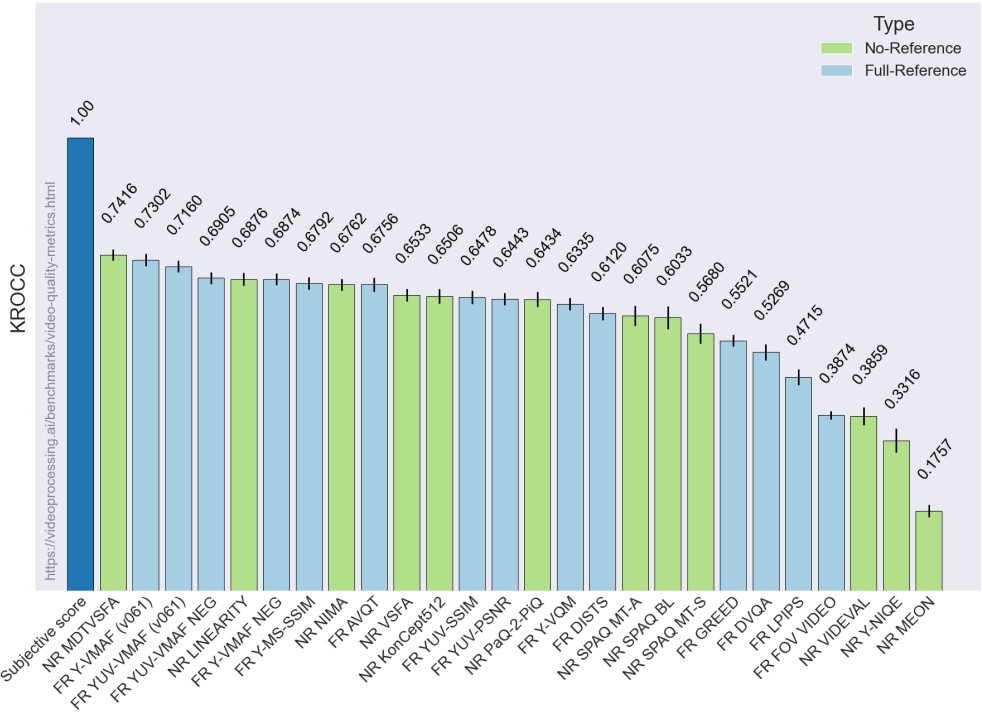

Figure 12: KROCC values for "Shaking" category.

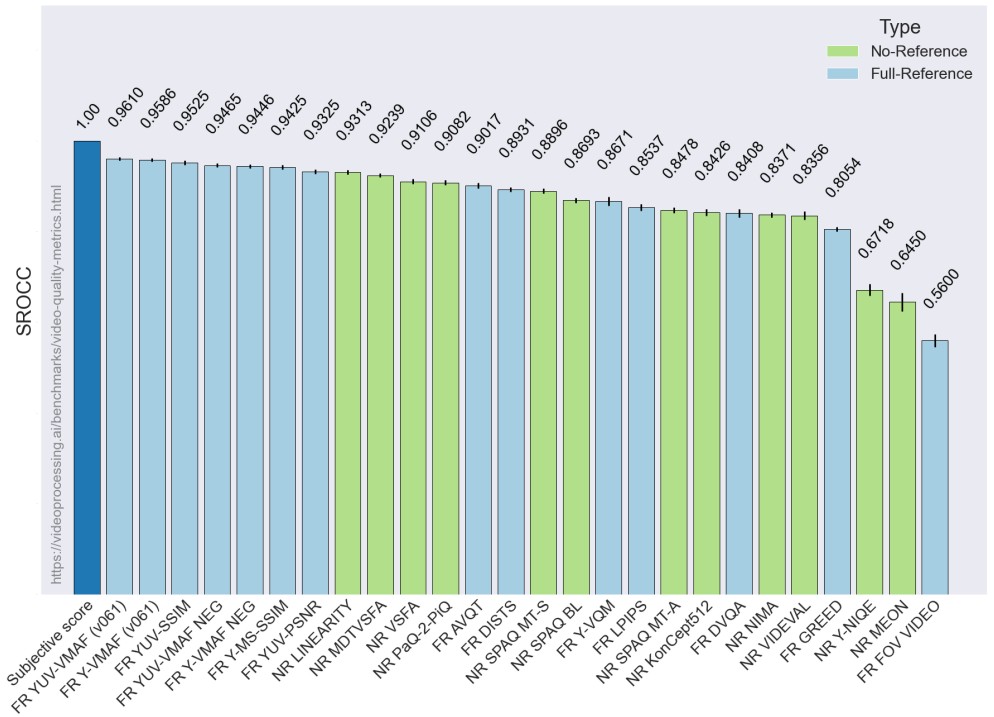

Figure 13: SROCC values for "Sports" category.

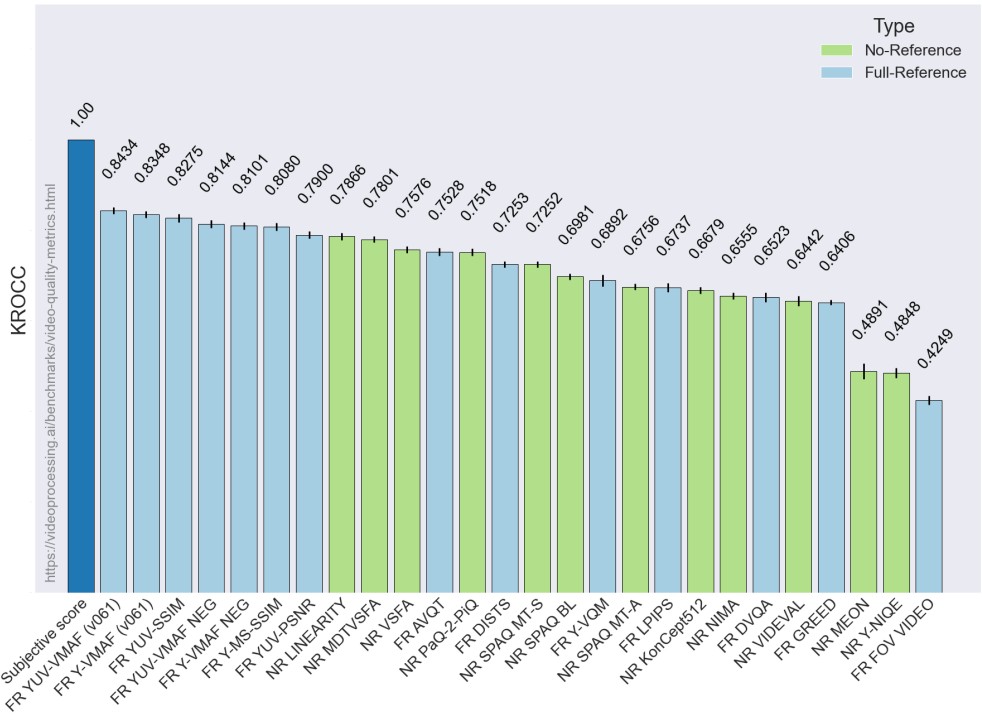

Figure 14: KROCC values for "Sports" category.

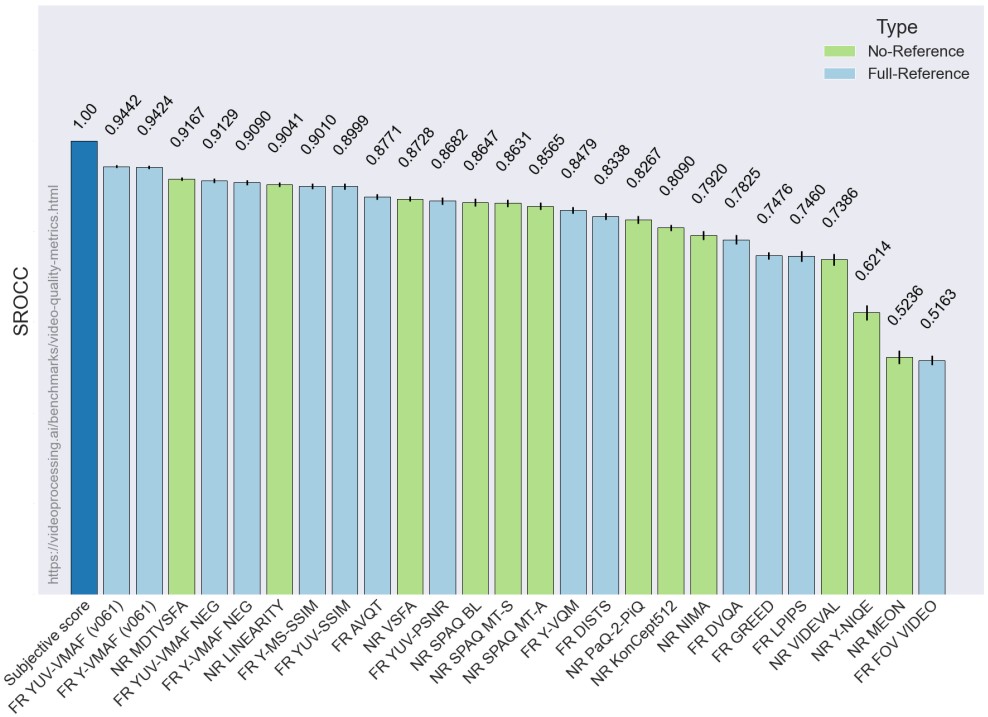

Figure 15: SROCC values for "Nature" category.

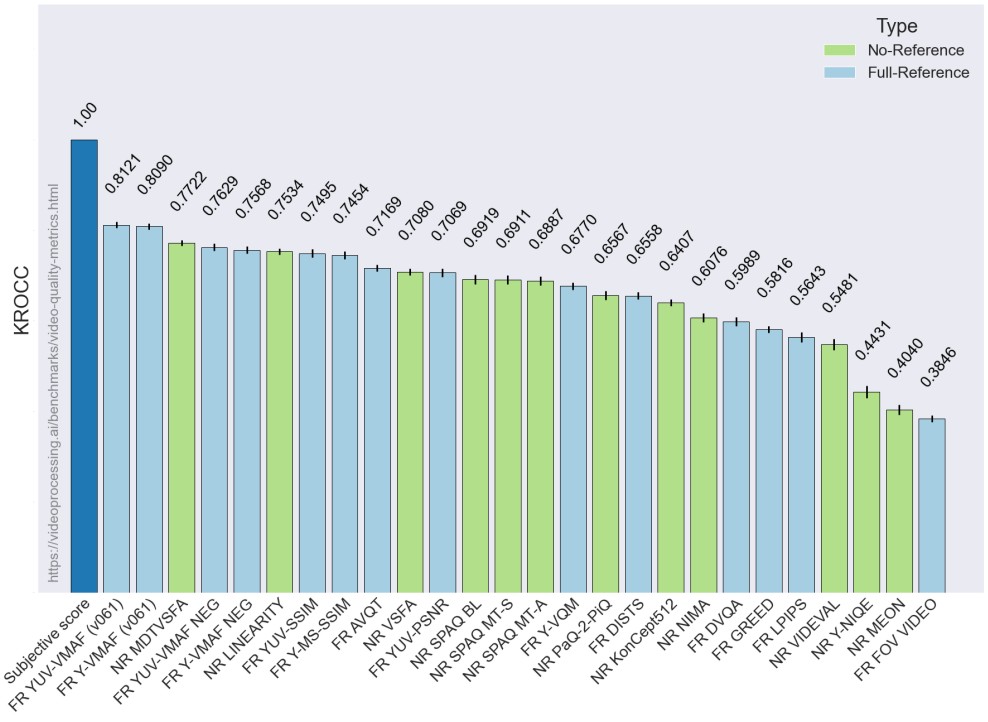

Figure 16: KROCC values for "Nature" category.

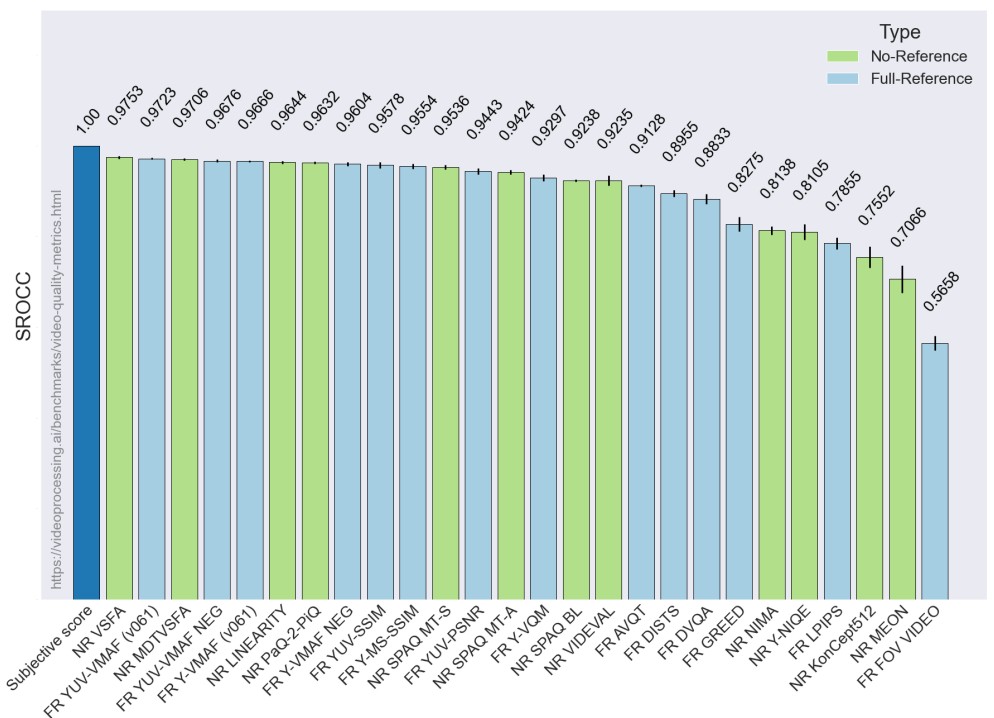

Figure 17: SROCC values for "Gaming / Animation" category.

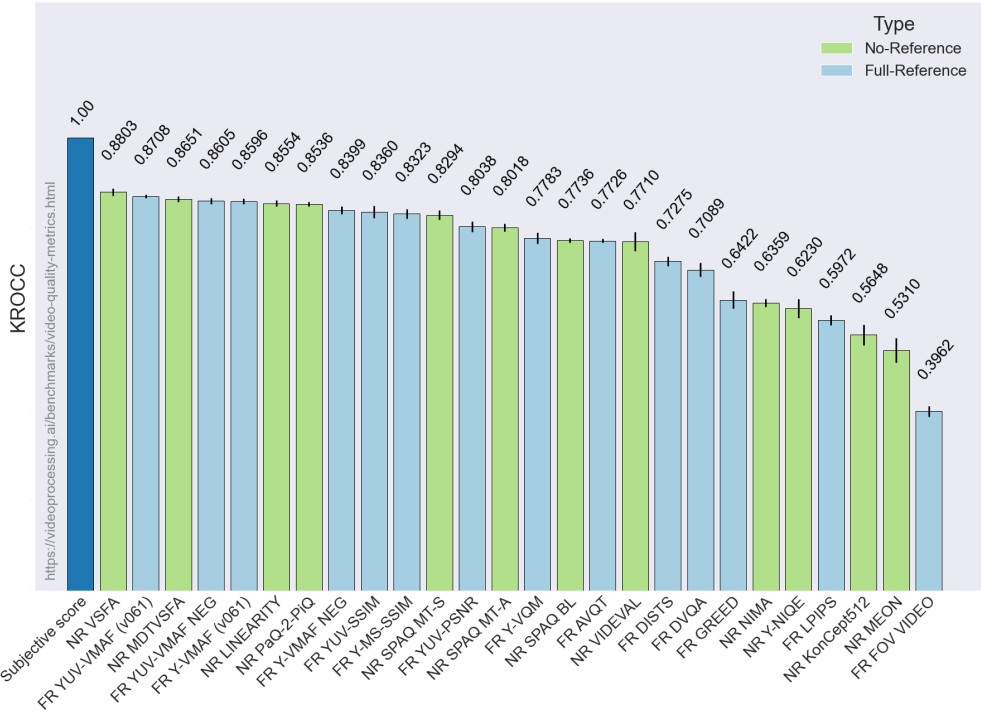

Figure 18: KROCC values for "Gaming / Animation" category.

## 1.5 Metrics Correlation for Different Compression Standards

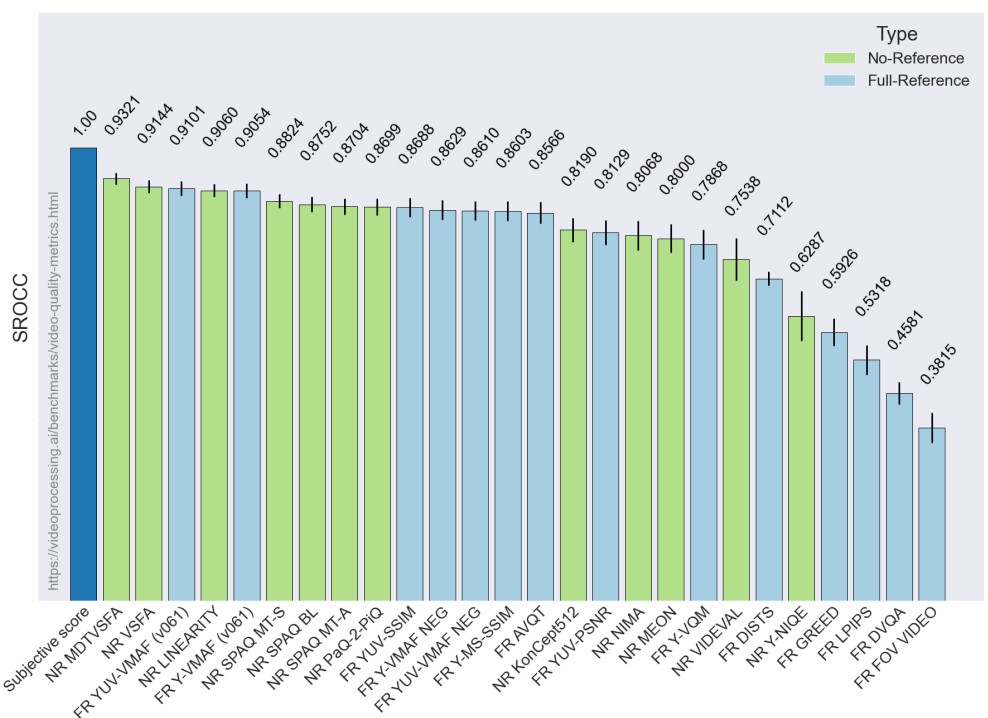

Figure 19: SROCC values for AV1 encoding standard.

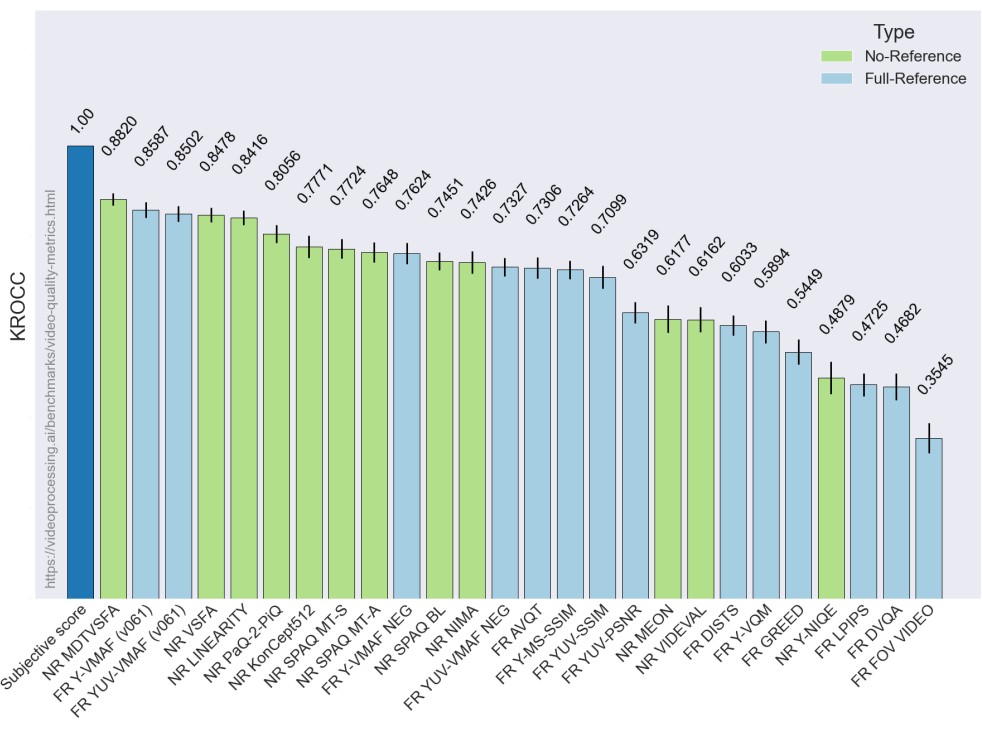

Figure 20: KROCC values for AV1 encoding standard.

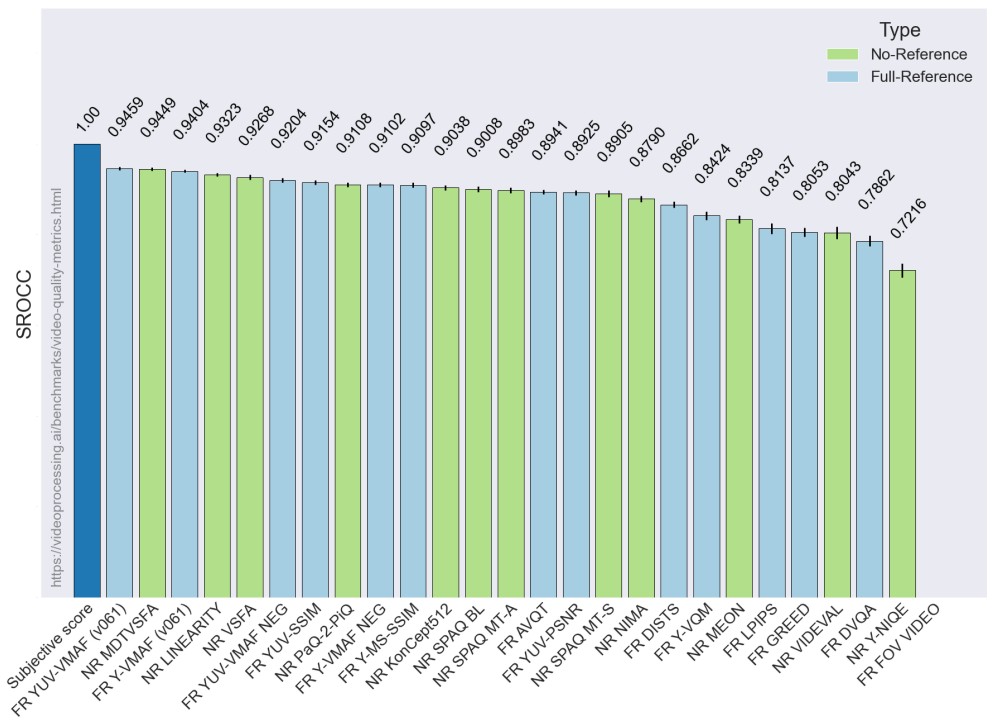

Figure 21: SROCC values for H.265 encoding standard.

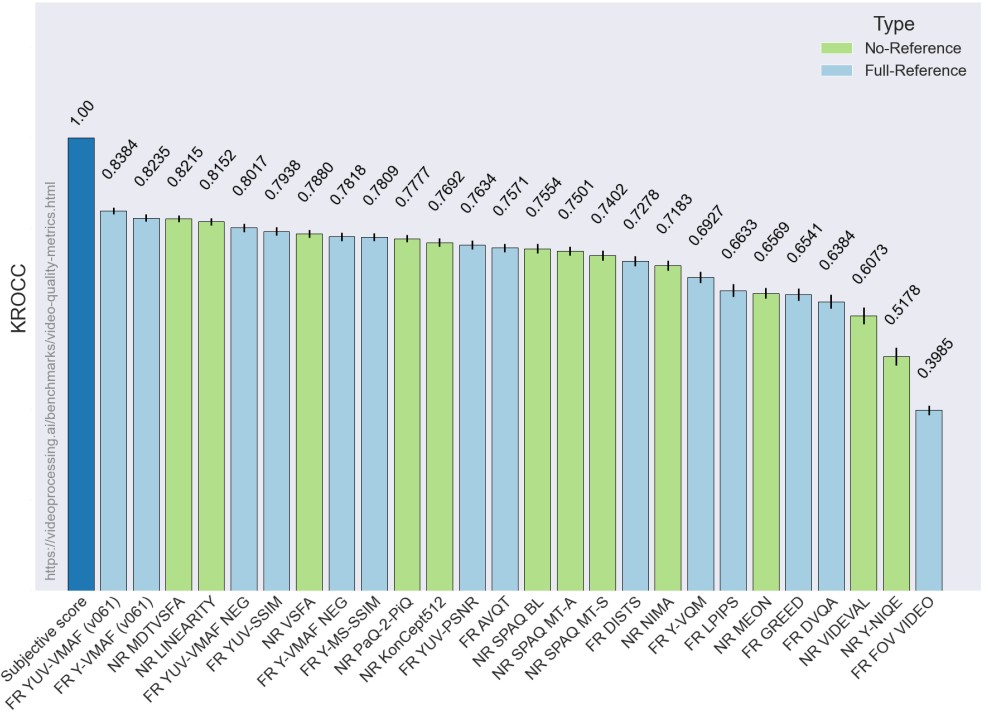

Figure 22: KROCC values for H.265 encoding standard.

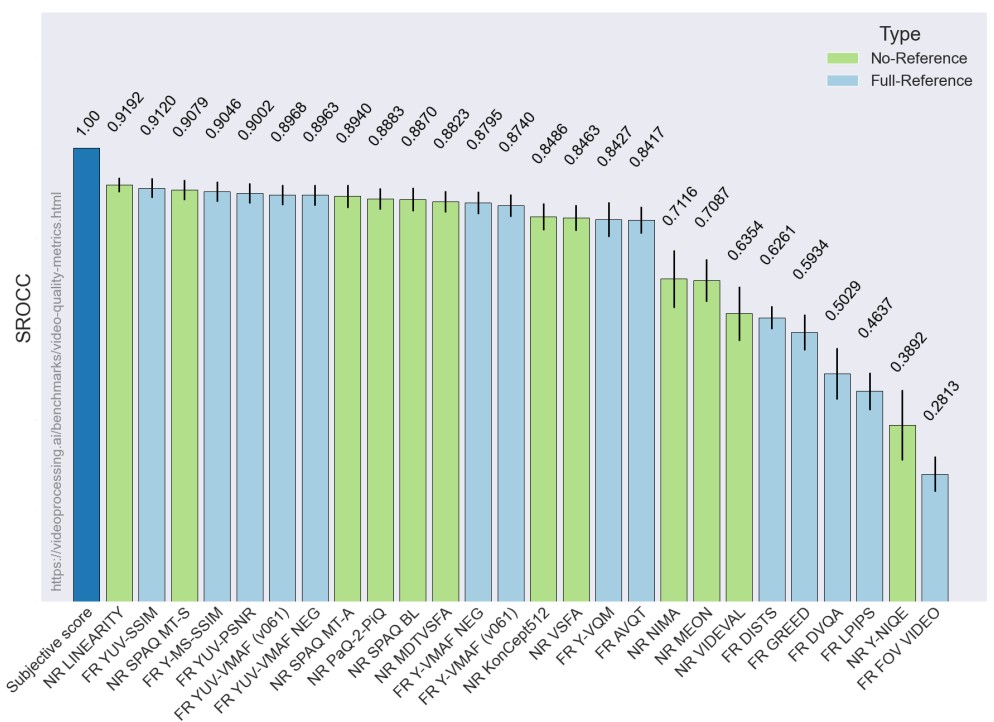

Figure 23: SROCC values for VVC encoding standard.

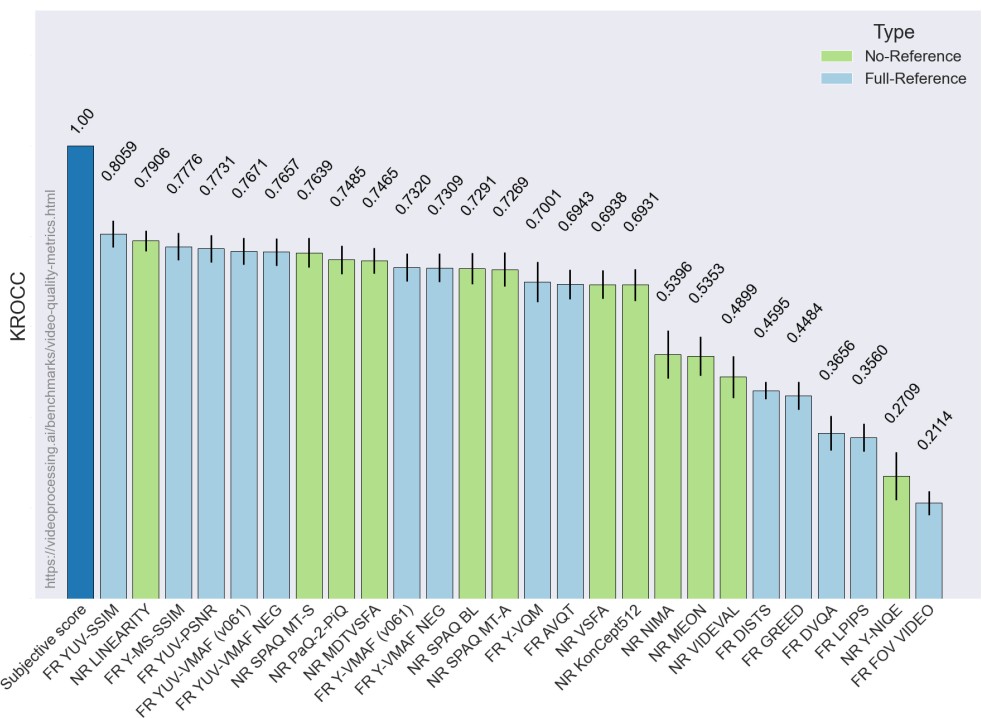

Figure 24: KROCC values for VVC encoding standard.

## 1.6 Wilcoxon Test

Table 1: Results of one-sided Wilcoxon rank-sum test performed on SROCC values for the methods compared above. For each pair of methods, the table shows values for 11 subsets of our dataset: in the first row are the entire dataset, Low Bitrate, and High Bitrate; in the second are H.265 Encoding, AV1 Encoding, and VVC Encoding; in the third are User-Generated Content, Shaking, and Sports; and in the fourth are Nature and Gaming/Animation. A value of 1 indicates the method for that row is statistically superior to the method for that column. A 0 value indicates the opposite: the method for that column is statistically better than the one for that row. A hyphen (-) indicates they are statistically indistinguishable.

### 1.7 How to Submit an Algorithm for Benchmarking

To submit an algorithm for benchmarking, send an email to vqa@videoprocessing.ai with the following information:

- Method name, which we will use in our benchmark
- Method-launch script with the following options (or their analogs):
    - -ref—path to reference video (for full-reference metrics)
    - -dist—path to distorted video
    - -output—path to algorithm's output
    - -t—threshold, if algorithm requires it
- Any other helpful information about the method (optional):
    - Desired parameters
    - Link to any papers about the method
    - Architectural characteristics

Our policy:

- We won't publish the results for a method without the submitter's permission.
- We share only the open part of our dataset; the rest is hidden.

## 2 Dataset Documentation

Here we provide documentation for our dataset in the common datasheets format [2].

### 2.1 Motivation

**For what purpose was the dataset created? Was there a specific task in mind? Was there a specific gap that needed to be filled?**

We produced the dataset to evaluate full- and no-reference video-quality metrics. To the best of our knowledge, it is the biggest compressed-video dataset that includes new encoding standards and subjective scores. Investigations of the performance of modern video- and image-quality metrics commonly employ videos compressed with older standards, such as AVC. Our goal was to create a dataset that uses numerous compression standards.

**Who created the dataset (for example, which team, research group) and on behalf of which entity (for example, company, institution, organization)?**

The dataset was a joint effort by Anastasia Antsiferova, Sergey Lavrushkin, Alexander Gushchin, Maksim Smirnov, Dmitriy Kulikov, Egor Sklyarov, Mikhail Erofeev, and Dmitriy Vatolin . The authors are researchers affiliated with the MSU Graphics and Media Lab.

**Who funded the creation of the dataset? If there is an associated grant, please provide the name of the grantor and the grant name and number.**

Part of the dataset was collected for the annual MSU Video Codecs Comparisons project [1].

The work received support through a grant for research centers in the field of artificial intelligence (agreement identifier 000000D730321P5Q0002, dated November 2, 2021, no. 70-2021-00142 with the Ivannikov Institute for System Programming of the Russian Academy of Sciences). Anastasia Antsiferova was supported by "Intellect", a noncommercial fund for science and educational development.

### 2.2 Composition

**What do the instances that comprise the dataset represent (for example, documents, photos, people, countries)? Are there multiple types of instances (for example, movies, users, and ratings; people and interactions between them; nodes and edges)?**

The instances represent video files with corresponding subjective scores. Each video was produced by compressing the original with a specific encoder.

**How many instances are there in total (of each type, if appropriate)?**

The dataset has a total of 2,486 compressed videos with corresponding subjective scores. The open part contains 1,022 videos; the hidden part contains 1,464.

**Does the dataset contain all possible instances or is it a sample (not necessarily random) of instances from a larger set? If the dataset is a sample, then what is the larger set? Is the sample representative of the larger set (for example, geographic coverage)?**

The dataset contains all generated instances. But since it also includes reference videos, the dataset can be extended using new video encoders. Original videos were chosen based on SI-TI characteristics; they cover most possible cases of spatial and temporal complexity.

**What data does each instance consist of? "Raw" data (for example, unprocessed text or images) or features?**

Each instance is a video file in .mp4 format. Each subjective score is a floating-point number.

**Is there a label or target associated with each instance?**

Each instance has a corresponding subjective score that represents relative video quality compared with other instances generated from the same original video.

**Is any information missing from individual instances?**

No.

**Are relationships between individual instances made explicit (for example, users' movie ratings, social network links)?**

We divided the videos into groups on the basis of a reference video, which is compressed using different encoders and bitrates to generate a group.

**Are there recommended data splits (for example, training, development/validation, testing)?**

We recommend splitting data according to the original videos. This way, all videos that are compressed representations of a given reference video will fall into the same group. Also, when obtaining the subjective scores, compressed versions of a given reference video were shown to crowdworkers, so correlations of video-quality-metric values with subjective scores apply only within a group.

**Are there any errors, sources of noise, or redundancies in the dataset?**

No.

**Is the dataset self-contained, or does it link to or otherwise rely on external resources (for example, websites, tweets, other datasets)?**

The dataset is self-contained.

**Does the dataset contain data that might be considered confidential (for example, data that is protected by legal privilege or by doctor-patient confidentiality, data that includes the content of individuals' non-public communications)?**

We allow free distribution of the dataset's open part. The hidden part contains confidential information that we must withhold.

**Does the dataset contain data that, if viewed directly, might be offensive, insulting, threatening, or might otherwise cause anxiety?**

No.

## 2.3 Collection Process

**How was the data associated with each instance acquired? Was the data directly observable (for example, raw text, movie ratings), reported by subjects (for example, survey responses), or indirectly inferred/derived from other data (for example, part-of-speech tags, model-based**

**guesses for age or language)? If the data was reported by subjects or indirectly inferred/derived from other data, was the data validated/verified?**

We formed the dataset using 36 reference clips chosen from more than 18,000 open-source high-bitrate videos (licensed under CCBY or CC0). They include recordings by professionals and amateurs. Almost half contain scene changes and high dynamism. Moreover, the ratio of synthetic to natural lightning is approximately 1:3.

Content types: nature, sports, humans close up, gameplay, music video, water or steam, and CGI. Effects and distortions: shaking, slow motion, grain/noise, overly dark/bright regions, macro shooting, captions (text), and extraneous objects on or near the camera lens. Such content diversity helps better simulate realistic conditions.

1. Resolution: 1,920×1,080—the most popular video resolution today (likely to increase over time)
2. Format: yuv420p.
3. FPS: 24, 25, 30, 39, 50, and 60.
4. Video duration: 10, 15 seconds (most cases).

We divided the video collection into 36 clusters using the K-means method. For each cluster, we randomly selected one to six candidate videos that were close to the cluster center and that had an appropriate license. From each set of candidates, we manually chose one video and attempted to include videos of different semantics in the final dataset.

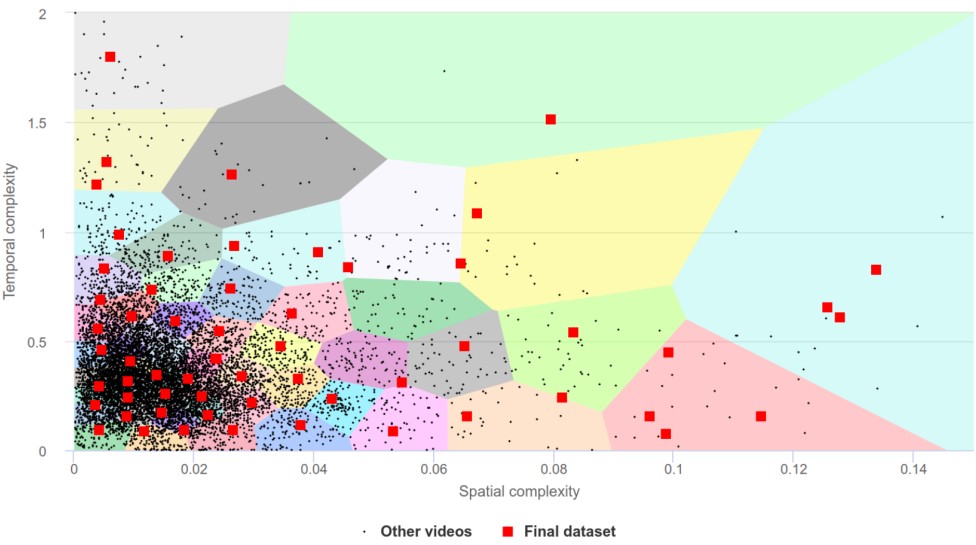

Figure 25: Selection of reference videos using the K-means clustering method.

Tab. 4 lists URLs for the original videos.

Details about the crowdsourced study:

- Screen resolutions were from 640×320 to 3,840×1,080. Tab. 3 shows the most popular ones.
- Participants were from 78 countries.
- Participant ages ranged from 18 to 85 with an average of 36. Fig. 26 shows the distribution.

**What mechanisms or procedures were used to collect the data (for example, hardware apparatuses or sensors, manual human curation, software programs, software APIs)? How were these mechanisms or procedures validated?**

The subjective assessment (labeling) involved pairwise comparisons using Subjectify.us. Each pair, which was shown to hired participants, consisted of two samples of the same test video encoded by

| Video URL | Author |
|---|---|
| https://vimeo.com/202290580 | Currently Unavailable |
| https://vimeo.com/87156909 | Al Caudullo Productions (https://vimeo.com/alfredcaudullo) |
| https://vimeo.com/192252473 | IA Film Group(https://vimeo.com/iafilmgroup) |
| https://vimeo.com/98238216 | AIE(https://vimeo.com/aieedu) |
| https://media.xiph.org/video/derf/ Name: crowd$_r$un | Xiph.org Video Test Media [derf's collection] |
| https://media.xiph.org/video/derf/ Name: tractor | Xiph.org Video Test Media [derf's collection] |
| https://vimeo.com/312309391 | krautmovies(https://vimeo.com/krautmovies) |
| https://vimeo.com/78721233 | SantaMarta and Astorga (https://vimeo.com/user5121153) |
| https://vimeo.com/259826267 | Animal Factory Films (https://vimeo.com/animalfactoryfilms) |
| https://media.withyoutube.com/(YoutubeUGCID:gaming_71a5) | Youtube UGC Dataset |
| https://media.withyoutube.com/(YoutubeUGCID:livemusic_3549) | Youtube UGC Dataset |
| https://media.withyoutube.com/(YoutubeUGCID:Vlog_1080P-35cd) | Youtube UGC Dataset |
| https://media.withyoutube.com/(YoutubeUGCID:CoverSong_1080p-5430) | Youtube UGC Dataset |
| https://media.withyoutube.com/(YoutubeUGCID:Vlog_1080P-52fe) | Youtube UGC Dataset |
| https://media.withyoutube.com/(YoutubeUGCID:Gaming_1080P-6db2) | Youtube UGC Dataset |
| https://media.withyoutube.com/(YoutubeUGCID:Vlog_1080P-5904) | Youtube UGC Dataset |
| https://media.withyoutube.com/(YoutubeUGCID:CoverSong_1080P-1b0c) | Youtube UGC Dataset |
| https://media.withyoutube.com/(YoutubeUGCID:Vlog_1080P-23cb) | Youtube UGC Dataset |
| https://media.withyoutube.com/(YoutubeUGCID:Vlog_1080P-4921) | Youtube UGC Dataset |
| https://media.withyoutube.com/(YoutubeUGCID:Vlog_1080P-21f5) | Youtube UGC Dataset |
| https://media.withyoutube.com/(YoutubeUGCID:Vlog_1080P-1df9) | Youtube UGC Dataset |
| https://media.withyoutube.com/(YoutubeUGCID:Vlog_1080P-2600) | Youtube UGC Dataset |
| https://vimeo.com/198898016 | Christopher Stoney (https://vimeo.com/cstoney) |
| https://vimeo.com/150329182 | Andrew Jones (/https://vimeo.com/jones3) |
| https://vimeo.com/163746200 | JTwo.tv (https://vimeo.com/jtwo) |
| https://vimeo.com/207154158#t=0 | niko (https://vimeo.com/igwana) |
| https://vimeo.com/204297495#t=140 | Scott (https://vimeo.com/slee1000) |
| https://vimeo.com/218791521#t=0 | Pyranha (https://vimeo.com/pyranhakayaks) |
| https://vimeo.com/280135236#t=347 | Currently Unavailable |
| https://vimeo.com/311282786#t=18 | Harold Aune (https://vimeo.com/haroldaune) |
| https://vimeo.com/245154516#t=42 | Rest Of My Family (https://vimeo.com/restofmyfamily) |
| https://vimeo.com/308829951#t=0 | Currently Unavailable |
| https://vimeo.com/189893327#t=173 | Currently Unavailable |
| https://vimeo.com/188799676#t=38 | Kona Bikes (https://vimeo.com/konaworld) |
| https://vimeo.com/130709443#t=49 | Keith W Roe (https://vimeo.com/user10596573) |
| https://vimeo.com/219044636#t=63 | george manolis (https://vimeo.com/user7033472) |

Table 2: URLs for original videos.

| Resolution | Number of users |
|---|---|
| 1366x768 | 10926 |
| 1920x1080 | 8421 |
| 1536x864 | 3397 |
| 1280x1024 | 2285 |
| 1600x900 | 2262 |
| 1440x900 | 1014 |
| 1280x720 | 984 |
| 1680x1050 | 677 |
| 1360x768 | 584 |
| 1280x800 | 420 |

Table 3: Most popular screen resolutions among crowdworkers.

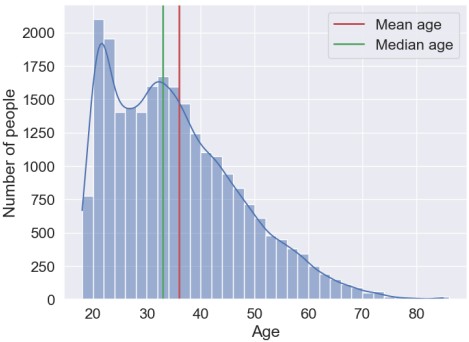

Figure 26: Age distribution of crowdworkers.

various codecs at various bitrates. For each pair, participants were asked to choose the one with better visual quality. They also had the option to play the videos again or to indicate the videos have equal quality. Fig. 27 depicts the subjective experiment's general process.

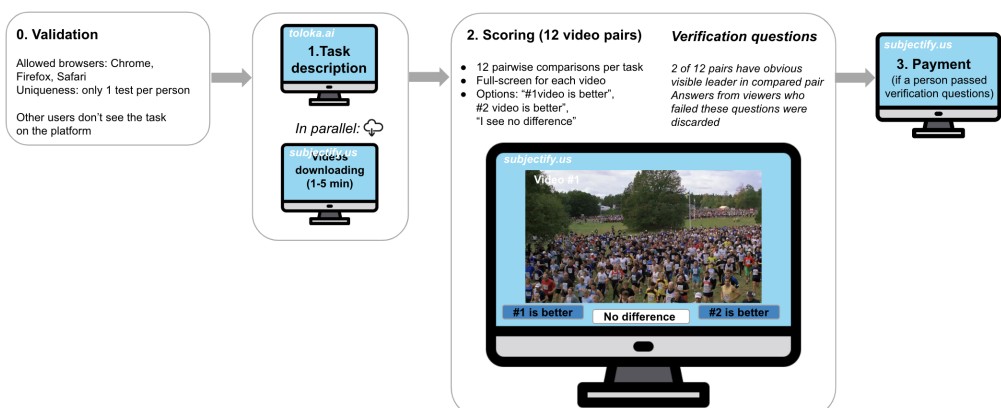

Figure 27: Subjective-assessment scheme.

Because the necessary number of pairwise comparisons grows exponentially with the number of source videos, we divided the dataset into five subsets by source videos and performed five comparisons. Each subset contained a group of source videos and their compressed versions. In each comparison, all possible pairs of compressed videos for one source video were generated and evaluated. Thus, only videos from the same source were in each pair. The comparison set also included source videos. Participants saw videos from each pair sequentially in full-screen mode. They were asked to choose the one with the best visual quality or indicate that the two are of the same quality. They also had an option to replay the videos. Their task was to evaluate a total of 12 video pairs, two of which had an obviously superior-quality option and served as verification questions. Responses from participants who failed either or both of these questions were discarded.

To increase the relevance of the results, we solicited at least 10 responses for each pair. In total, we were able to collect 766,362 valid answers from nearly 11,000 individuals.

After applying the Bradley-Terry model to a table of pairwise ranks, we received subjective scores that are consistent within each group of videos compressed from a single reference video.

**If the dataset is a sample from a larger set, what was the sampling strategy (for example, deterministic, probabilistic with specific sampling probabilities)?**

The open part is a sample of our full dataset. It includes all nonconfidential material. The hidden part contains confidential information and cannot be published. We employ the former only for our benchmark testing to allow a more objective comparison of future algorithms. This approach may prevent learning-based methods from training on the entire dataset, avoiding overfitting and, thus, incorrect results. The full dataset is not a sample of a larger set.

**Who was involved in the data collection process (for example, students, crowdworkers, contractors) and how were they compensated (for example, how much were crowdworkers paid)?**

The subjective scores were collected through the `www.subjectify.us` crowdsourcing platform. The average payment to crowdworkers per pair of sequences was $0.05. We estimate the overall cost of the subjective tests was $15,000.

**Over what timeframe was the data collected? Does this timeframe match the creation timeframe of the data associated with the instances (for example, recent crawl of old news articles)?**

Collection of the dataset took place over four years, but the timeframe is immaterial in our case.

**Were any ethical review processes conducted (for example, by an institutional review board)?**

No, such processes were unnecessary in our case.

## 2.4 Preprocessing/Cleaning/Labeling

**Was any preprocessing/cleaning/labeling of the data done (for example, discretization or bucketing, tokenization, part-of-speech tagging, SIFT feature extraction, removal of instances, processing of missing values)?**

We encoded all samples using x264 with a constant quantization parameter and then calculated the temporal and spatial complexity of each scene. In our definition, spatial complexity is the average size of the I-frame normalized to the sample's uncompressed frame size, while temporal complexity is the average size of the P-frame divided by the average size of the I-frame. Eventually, we added another preprocessing step to unify the chroma subsampling of videos, which affected evaluation complexity . All videos were converted to a YUV 4:2:0 chroma subsample.

For the open part of the dataset, we used 13 codecs implementing five compression standards (H.264, AV1, H.265, VVC, etc.) to encode the original videos. Each video had three target compression bitrates: 1,000 Kbps, 2,000 Kbps, and 4,000 Kbps. Each also had different real-life encoding modes: constant quality (CRF) and variable bitrate (VBR). Our chosen range of bitrates simplifies the subjective comparison, since video quality is more difficult to distinguish visually at high bitrates.

To save repository space and prevent problems with nonstandard decoders, we transcoded YUV files to MP4 using lossless x265 encoding and the following command:

```
ffmpeg -f rawvideo -vcodec rawvideo -s {width}x{height}
-r {FPS} -pix_fmt yuv420p -i {video name}.yuv -c:v libx265
-x265-params "lossless=1:qp=0" -t {hours:minutes:seconds}
-vsync 0 {video name}.mp4
```

To decode videos back to YUV, we employed this command:

```
ffmpeg -i {video name}.mp4 -pix_fmt yuv420p -vcodec rawvideo
-f rawvideo {video name}.yuv
```

Because Bradley-Terry scores were calculated only for encoded streams with the same reference video, correlations were calculated separately for each reference video (and corresponding encoded streams). To compute a single correlation for a whole dataset, we used the Fisher Z-transform to average group correlations weighted proportionally to group size. The quality-control pairs consisted of test videos compressed by the x264 encoder at 1 Mbps and 4 Mbps; they protect the comparison against random answers and bots. Responses from participants who failed in these cases were excluded.

**Was the "raw" data saved in addition to the preprocessed/cleaned/labeled data (for example, to support unanticipated future uses)?**

If we interpret "raw" data as streams encoded before decoding them back to YUV and transcoding with x265, the answer is no. But we share ground-truth (or reference) videos, which are necessary when calculating full-reference metrics.

**Is the software that was used to preprocess/clean/label the data available?**

We obtained some metric scores (VMAF, VQM, PSNR, SSIM, etc.) through the VQMT, available at www.compression.ru/video/quality_measure/vqmt_download.html. Also, all data was labeled through the Subjectify.us crowdsourcing service, which is at www.subjectify.us.

## 2.5 Uses

**Has the dataset been used for any tasks already?**

The dataset has served in two separate projects:

1. Measurement of video-quality metrics for distorted videos along with calculation of the correlation between these scores and subjective scores (video-quality-metric benchmark).

2. Comparing the effects (artifacts) of different compression standards and codecs (codec-comparison project).

The second project held annual comparisons using different datasets, from which we assembled the dataset in our paper.

**Is there a repository that links to any or all papers or systems that use the dataset?**

Video-quality benchmark is available through `www.videoprocessing.ai/benchmarks/video-quality-metrics.html` and the codec-comparison project through `www.compression.ru/video/codec_comparison/index_en.html`.

**What (other) tasks could the dataset be used for?**

The open part of the dataset can be used to train or validate a new metric and determine whether it can reliably meet most compression needs. Our video-quality-metric benchmark uses the hidden part to test submitted models.

**Is there anything about the composition of the dataset or the way it was collected and preprocessed/cleaned/labeled that might impact future uses? For example, is there anything that a dataset consumer might need to know to avoid uses that could result in unfair treatment of individuals or groups (for example, stereotyping, quality of service issues) or other risks or harms (for example, legal risks, financial harms)? Is there anything a dataset consumer could do to mitigate these risks or harms?**

One preprocessing stage is transcoding through the H.265 standard, which can affect bitstream-based-model performance. In addition, consumers cannot use subjective scores to directly compare distorted videos compressed from different original streams because of the subjective-evaluation procedure we employed for the dataset.

**Are there tasks for which the dataset should not be used?**

No.

## 2.6 Distribution

**Will the dataset be distributed to third parties outside of the entity (for example, company, institution, organization) on behalf of which the dataset was created?**

The open part of the dataset is available to everyone. The hidden part is only available to benchmark-support personnel for testing metric performance.

**How will the dataset be distributed (for example, tarball on website, API, GitHub)? Does the dataset have a digital object identifier (DOI)?**

The open part of the dataset is accessible through `https://calypso.gml-team.ru:5001/sharing/lxSWi6vtg` using the password c943=R3/tJwVV%P%. The benchmark is at `www.videoprocessing.ai/benchmarks/video-quality-metrics.html`

**When will the dataset be distributed?**

We have already released the dataset as test data for our benchmark, and we are now accepting submissions.

**Will the dataset be distributed under a copyright or other intellectual property (IP) license, and/or under applicable terms of use (ToU)?**

The dataset is available under a CCBY license.

**Have any third parties imposed IP-based or other restrictions on the data associated with the instances?**

All original videos have CCBY and CC0 licenses.

**Do any export controls or other regulatory restrictions apply to the dataset or to individual instances?**

No.

## 2.7 Maintenance

**Who will be supporting/hosting/maintaining the dataset?**

The CMC MSU Graphics and Media Lab hosts the dataset. The team that works with codecs and video-quality assessment methods maintains it. Also, the authors of this paper support the video-quality-metric benchmark.

**How can the owner/curator/manager of the dataset be contacted (for example, email address)?**

Contact our team at vqa@videoprocessing.ai.

**Is there an erratum?**

No. The dataset has remained unchanged since the benchmark's release.

**Will the dataset be updated (for example, to correct labeling errors, add new instances, delete instances)? If so, please describe how often, by whom, and how updates will be communicated to dataset consumers (for example, mailing list, GitHub)?**

We are planning to extend the dataset to ensure benchmark results with the highest statistical credibility. Such updates will be rare, as they involve subjective evaluation—a time-consuming task that requires extensive preparation. Also, we understand the problems that consumers can face during updates. But after updates become public, they will receive notification primarily through the mailing list, and all the new information will be on the benchmark website.

**If the dataset relates to people, are there applicable limits on the retention of the data associated with the instances (for example, were the individuals in question told that their data would be retained for a fixed period of time and then deleted)?**

No.

**Will older versions of the dataset continue to be supported/hosted/maintained?**

We do not intend to create a version history, as every relevant edition of the dataset will include all previous editions.

**If others want to extend/augment/build on/contribute to the dataset, is there a mechanism for them to do so? Will these contributions be validated/verified? If not, why not? Is there a process for communicating/distributing these contributions to dataset consumers?**

We encourage everyone to share their ideas on extending our dataset to cover more compression cases and provide more-reliable results. Our method of subjective quality evaluation, however, is set; we recommend consumers contacting us by vqa@videoprocessing.ai to coordinate subjective quality evaluation. Moreover, we can cover the costs of subjective scoring for videos that we find relevant to our research.

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

| Encoder name and version | Encoding commands |
| --- | --- |
| x265
v.3.0+1-ed72af837053 | "x265.exe –input %SOURCE_FILE% –input-res %WIDTH%x%HEIGHT% –fps %FPS%
-p veryslow –bitrate %BITRATE_KBPS% –psnr –ssim –tune=ssim -o %TARGET_FILE%" |
| x264
v.r2969-d4099dd | "x264 –preset placebo –me umh –merange 32 –keyint infinite –tune ssim –pass 1
–bitrate %BITRATE_KBPS% %SOURCE_FILE% –input-res %WIDTH%x%HEIGHT% –fps %FPS% -o NUL
x264 –preset placebo –me umh –merange 32 –keyint infinite
–tune ssim –pass 2 –bitrate %BITRATE_KBPS% %SOURCE_FILE%
–input-res %WIDTH%x%HEIGHT% –fps %FPS% -o %TARGET_FILE%" |
| vp9
v1.8.0-424-ge50f4e411 | "vpxenc.exe %SOURCE_FILE% -o %TARGET_FILE% –codec=vp9 –good -p 2 –target-bitrate=%BITRATE_KBPS%
–profile=0 –lag-in-frames=25 –min-q=0 –max-q=63 –auto-alt-ref=1 –passes=2 –kf-max-dist=9999
–kf-min-dist=0 –drop-frame=0 –static-thresh=0 –bias-pct=50 –minsection-pct=0 –maxsection-pct=2000 –arnr-maxframes=7
–arnr-strength=5 –sharpness=0 –undershoot-pct=100 –overshoot-pct=100 –frame-parallel=0 –row-mt=1
–width=%WIDTH% –height=%HEIGHT% –fps=%FPS_NUM%/%FPS_DENOM%
–tile-columns=0 –cpu-used=0 –threads=32" |
| xin265
1.0 | "xin265_enc.exe -w %WIDTH% -h %HEIGHT% -f %FPS% -b %BITRATE_BPS%
-o %TARGET_FILE% -p 4 -i %SOURCE_FILE%" |
| x264
r3018-db0d417 | "x264.exe –preset placebo –me umh –merange 32 –keyint infinite –tune ssim
–crf {TARGET_RC} {SOURCE_FILE} –input-res {WIDTH}x{HEIGHT} –fps FPS -o TARGET_FILE"
"x264.exe –preset slow –subme 9 –bframes 8 –me umh –keyint infinite –tune ssim
–crf TARGET_RC SOURCE_FILE –input-res WIDTHxHEIGHT –fps FPS -o TARGET_FILE" |
| x265
2020-04-13 | "x265.exe –tune ssim –preset medium –crf TARGET_RC SOURCE_FILE -o TARGET_FILE
–input-res WIDTHxHEIGHT –fps FPS"
"x265.exe –tune ssim –pass 1 –preset veryslow –crf TARGET_RC
SOURCE_FILE -o TARGET_FILE –input-res WIDTHxHEIGHT –fps FPS –vbv-bufsize 12000 –vbv-maxrate 12000
x265.exe –tune ssim –pass 2 –preset veryslow –crf TARGET_RC SOURCE_FILE -o TARGET_FILE
–input-res WIDTHxHEIGHT –fps FPS –vbv-bufsize 12000 –vbv-maxrate 12000"
"x265.exe –preset veryslow –tune ssim –rskip 0 –ref 5 –merange 92
–rc-lookahead 50 –crf TARGET_RC SOURCE_FILE -o TARGET_FILE –input-res WIDTHxHEIGHT –fps FPS" |
| rav1e | "rav1e.exe –threads 12 –tiles 8 –min-keyint 25 –keyint 250 –speed 3
–quantizer TARGET_RC -o TARGET_FILE.ivf SOURCE_FILE" |
| SVT-AV1
v0.8.3 | "SvtAv1EncApp.exe -i %SOURCE_FILE% -w %WIDTH% -h %HEIGHT% –fps %FPS% –rc 0
-q %BITRATE_KBPS% –preset 3 -b %TARGET_FILE%" |
| SVT-VP9
v0.2.0 | "SvtVp9EncApp.exe -i %SOURCE_{FILE}% − w%WIDTH% − h%HEIGHT% − fps%FPS% − rc0
-q %BITRATE_KBPS% -enc-mode 0 -b %TARGET_FILE%" |
| SVT-HEVC
v1.4.3 | "SvtHevcEncApp.exe -i %SOURCE_FILE% -w %WIDTH% -h %HEIGHT% -fps %FPS% -rc 0
-q %BITRATE_KBPS% -encMode 0 -b %TARGET_FILE%" |
| x264
0.160.3000 33f9e14 | "x264-r3018-db0d417.exe –preset placebo –me umh
–merange 32 –keyint infinite –tune ssim –crf %BITRATE_KBPS% %SOURCE_FILE%
–input-res %WIDTH%x%HEIGHT% –fps %FPS% -o %TARGET_FILE%" |
| x265
3.3+21-6bb2d88029c2 | "x265-64bit-8bit-2020-04-13.exe –tune ssim –preset ultrafast –crf %BITRATE_KBPS%
%SOURCE_FILE% -o %TARGET_FILE% –input-res %WIDTH%x%HEIGHT% –fps %FPS%" |
| SVT-HEVC
v1.5.1 | "SvtHevcEncApp.exe -i SOURCE_FILE -w WIDTH -h HEIGHT -fps FPS
-rc 1 -tbr TARGET_BITRATE -encMode 0 -b TARGET_FILE"
"SvtHevcEncApp.exe -i SOURCE_FILE -w WIDTH -h HEIGHT -fps FPS
-rc 1 -tbr TARGET_BITRATE -encMode 5 -b TARGET_FILE" |
| SVT-VP9
v0.3.0 | "SvtVp9EncApp.exe -i SOURCE_FILE -w WIDTH -h HEIGHT -fps FPS
-rc 1 -tbr TARGET_BITRATE -enc-mode 0 -b TARGET_FILE"
"SvtVp9EncApp.exe -i SOURCE_FILE -w WIDTH -h HEIGHT -fps FPS
-rc 1 -tbr TARGET_BITRATE -enc-mode 5 -b TARGET_FILE" |
| x264
r3065-ae03d92 | "x264-r3065-ae03d92.exe –preset placebo –me umh –merange 32
–keyint infinite –tune ssim –bitrate TARGET_BITRATE
SOURCE_FILE –input-res WIDTHxHEIGHT –fps FPS -o TARGET_FILE"
"SvtVp9EncApp.exe -i SOURCE_FILE -w WIDTH -h HEIGHT -fps FPS
-rc 1 -tbr TARGET_BITRATE -enc-mode 5 -b TARGET_FILE" |
| x265
3.5+1-ce882936d | "x265-8bit.exe –input SOURCE_FILE –input-res WIDTHxHEIGHT –fps FPS
–bitrate TARGET_BITRATE –preset medium –limit-sao –limit-ref 0 -o TARGET_FILE –psnr –ssim –tune=ssim"
"x265-8bit.exe –input SOURCE_FILE –input-res WIDTHxHEIGHT –fps FPS
–crf TARGET_RC –vbv-bufsize 12000 –vbv-maxrate 12000 –preset veryslow –rskip 0
–ref 5 –merange 92 –rc-lookahead 50 -o TARGET_FILE –psnr –ssim –tune=ssim" |
| x265
3.5+1-f0c1022b6 | "x265.exe –input SOURCE_FILE –input-res WIDTHxHEIGHT –fps FPS
–preset medium –bitrate TARGET_BITRATE –psnr –ssim –tune=ssim -o TARGET_FILE –sao –limit-sao –ctu 64 –ref 3 –limit-ref 0"
"x265.exe –input SOURCE_FILE –input-res WIDTHxHEIGHT –fps FPS
-p placebo –bitrate TARGET_BITRATE –psnr –ssim –tune=ssim -o TARGET_FILE" |
| vvenc
v1.0.0 | "./vvencapp –preset fast -i SOURCE_FILE -s WIDTHxHEIGHT -r FPS
–bitrate TARGET_BITRATE -o TARGET_FILE" |
| xin
v1.0 | "xin26x_test.exe -o TARGET_FILE -i SOURCE_FILE -w WIDTH -h HEIGHT
-f FPS -n FRAMES_NUM -r 1 -b TARGET_BITRATE -p 3 -a 0"
"xin26x_test.exe -o TARGET_FILE -i SOURCE_FILE -w WIDTH -h HEIGHT
-f FPS -n FRAMES_NUM -r 1 -b TARGET_BITRATE -p 5 –intranxn 1 –transformskip 1 -a 0" |
| xin vvc
v1.0 | "xin26x_test.exe -o TARGET_FILE -i SOURCE_FILE -w WIDTH -h HEIGHT
-f FPS -n FRAMES_NUM -r 1 -b TARGET_BITRATE -p 1 -a 2"
"xin26x_test.exe -o TARGET_FILE -i SOURCE_FILE -w WIDTH -h HEIGHT
-f FPS -n FRAMES_NUM -r 1 -b TARGET_BITRATE -p 4 -a 2" |
| rav1e
0.5.0-alpha (p20210518) | "rav1e-ch.exe –threads 16 –tiles 8 –slots 2 –min-keyint 25
–keyint 250 –speed 3 –quantizer TARGET_RC –frame-rate FPS_NUM –time-scale FPS_DENOM
-o TARGET_FILE.ivf SOURCE_FILE" |

Table 4: Encoding commands used for creating compressed videos.