# OpenReview forum: "Video compression dataset and benchmark of learning-based video-quality metrics"
_NeurIPS.cc/2022/Track/Datasets_and_Benchmarks — NeurIPS 2022 Datasets and Benchmarks _

### Official Review · Reviewer_Wdzx · 2022-07-17
**review for “Video compression dataset and benchmark of learning-based video-quality metrics”**

**Rating:** 7
**Confidence:** 5
**Correctness:** Yes
**Clarity:** Yes

**Strengths:**

(1) First, because of the high cost and complexity of implementation, most previous video datasets with scoring is small-scale and performed at the laboratory scale. However, in this work, the authors not only use extremely large scale crowdsourcing to perform the video scoring annotation work, but also set up a reasonable mechanism to detect mislabeling.
(2) Meanwhile, in the process of video data selection, the authors adopt the K-Means clustering method, thus dividing the videos into 87 classes based on spatio-temporal complexity, which greatly ensures the diversity of the dataset.

**Weaknesses:**

It is a great pity that we do not have a full picture of the dataset due to the authors' choice of partial disclosure.

**Additional Feedback:**

No

**Documentation:**

Unfortunately, the author intends to publish only 40% of the dataset.

**Relation To Prior Work:**

Yes

**Summary And Contributions:**

There are many video and photo quality metrics. However, most of them focus on resolution and quality subsets, temporal pooling, and computational-complexity-evaluation, but ignore video compression quality, which has become increasingly important in today's network era. As a result, the authors present the largest compressed video dataset, which include new coding standards and subjective ratings.

---

> ### Author Response · Authors · 2022-08-17
> **Response to Reviewer Wdzx**
>
> Thank you for your review and feedback!
> We chose a partial disclosure strategy to keep the benchmark unbiased and prevent methods from overfitting on open data. Also, we collected the dataset within MSU Video Codecs Comparisons project, where some of the participants encoders are nor open-source and so their encoded streams can not be shared.
> We will enlarge the collection of the benchmark with more videos and encoders, so the size of the open part will be increased.

---

### Official Review · Reviewer_YC5y · 2022-07-21
**A well-positioned and thorough study of video quality assessment**

**Rating:** 8
**Confidence:** 3
**Clarity:** The paper is clear and easy to follow.

**Strengths:**

Good paper overall!

1) The study is well-positioned: the drawbacks of the existing evaluation protocols are stated clearly, and are being addressed with the novel benchmark.

2) The large amount of data was collected in this study. The data acquisition procedure is well-structured and expounded clearly. I personally appreciate the authors verified the crowd-sourced measurements via additional questions. Noisy and misleading samples seem to be quite common in the open-source datasets due to inaccurate data acquisition protocols, so I find the verification part extremely important.

3) The experiments are comprehensive and persuading, since a lot of metrics were tested and many aspects of quality assessment are covered. The amount of the quantitative results presented in the forms of tables, histograms, and plots makes a good impression.

4) The authors provide a theoretical explanation of the empirical results. The evaluated metrics are discussed, their limitations and strengths are explicated; for some metrics, the most suitable usage scenarios are proposed (e.g., gaming/animation videos, or sports videos).

**Weaknesses:**

I am a bit confused with some aspects of the paper, and would be grateful for some additional comments from the authors.

1) In 3.1, it is stated that any metrics that have already shown low correlations in other studies are excluded from the evaluation. What are these metrics? Could you please provide references for the studies proving such metrics to be inadequate?

2) In 3.2, the authors claim the video quality for FullHD resolution is more difficult to distinguish visually during subjective tests. Is this conclusion based on a common sense or is it a result of an empirical study? It seems that the major advantage of the proposed benchmark is that it is modern and uses up-to-date codecs. However, it also seems that FullHD will remain a popular format within the next few years, so it is important to study them thoroughly.

3) It is unclear to me, do the open and the hidden parts of the dataset differ statistically? I would like to see data statistics calculated for the both parts separately.

4) The content types are UGC, gaming/animation, sports, nature, shaking. Could the authors shed some light on these categories& Are them mutually exclusive, e.g., an UGC video cannot be classified as shaking, or cannot depict nature? How these categories are assigned?

**Additional Feedback:**

See Weaknesses and Documentation.

**Correctness:**

The submission is a video dataset and a video quality assessment benchmark associated with the newly collected data. The metrics are common and widely-used. The data acquisition protocol relies on the statistical rules and includes a verification step. Overall, both the data acquisition and evaluation procedures seem to be correct and appropriate.

**Documentation:**

There is a website dedicated to the benchmark with a detailed information about the data collection and evaluation methodology, yet the data structure is not described. Currently, the dataset is available via a secure link. The authors have also presented a leaderboard for future submissions. However, these submissions are to be evaluated non-automatically but in a semi-manual mode: it makes the evaluation procedure less transparent and seems to be unconvenient for participants.

**Ethics:**

The hidden part of the dataset is claimed to be confidential; however, it is not to be open-sourced. According to the datasheet provided in the supplementary materials, the videos were collected from the users all around the world from almost 80 countries, aged from 18 to 85, so I believe the data acquisition protocol aims to facilitate diversity and provide a fair and comprehensive social representation.

**Relation To Prior Work:**

The authors provide a quantitave summary of benchmarks on video quality measurement in Tab. 2 and explain their motivation. As stated, the most recent video encoders produce artifacts, which are not considered by the quality metrics, making them irrelevant. The proposed benchmark addresses this issue by including videos compressed via multiple modern video encoders.

**Summary And Contributions:**

This work is dedicated to video quality assessment via trainable algorithms, focusing on measuring video quality after compression. The existing benchmarks contain videos compressed using outdated standards, thus they might seem irrelevant for evaluating modern algorithms. This paper introduces a new video quality benchmark for measuring the quality of videos encoded via nine actual standards. The authors acquired videos encoded using different standards, and performed a user study on subjective video quality using crowd-sourcing platforms. The collected dataset was used to evaluate a large number of no-reference and full-reference metrics. According to the conducted empirical study, the best full-reference metric is VMAF, which is highly correlated with the subjective quality, while the no-reference MDTVSFA metric is almost on par with full-reference metrics, yet being impractical due to computational complexity.

---

> ### Author Response · Authors · 2022-08-17
> **Response to Reviewer YC5y**
>
> Thank you for your detailed review! Answering your questions:
>
> 1. We added a remark into the paper that BRISQUE and VIIDEO are two well-known metrics that commonly appear in other comparisons and showed low correlations in at least three studies (Z. Ying et al. Patch-vq: patching up the video quality problem; D. Li et al. Unified quality assessment of in-the-wild videos with mixed datasets training; Y. Li et al. User-generated video quality assessment: A subjective and objective study). There are more metrics that showed low correlations in less number of studies that also received a lower priority status for evaluation. However, we plan to add as many metrics as possible, including BRISQUE and VIIDEO as there is still a chance that they perform well in some of our compression categories.
> 2. We agree that nowadays and within the next years FullHD format will likely remain the most popular format. There was a phrasing issue in the paper that we fixed. Our point was about appropriate encoding bitrates for FullHD format. We added the reference that major streaming video services recommend 4500-8000 kbps as an upper bitrate bound for FullHD encoding. However, according to our experience, the usage of bitrates higher than 6000 for FullHD makes subjective comparisons very difficult. It becomes very hard (and sometimes nearly impossible) to receive enough answers from crowdworkers to make confidence intervals small (the quality is too high and indistinguishable). Thus, we used 1,000 kbps, 2,000 kbps, and 4,000 kbps as target encoding bitrates in our dataset.
> 3. The difference in statistics that we tracked for open and hidden parts is small. We added the illustration of bitrate distribution in Figure 2 separately for open and hidden parts (the form of the distribution is almost the same, open part contains more extremely high bitrates and hidden part contains higher number of commonly used bitrates about 1,000 kbps). We also presented the difference in PSNR range and distribution in Figure 3 (open and hidden parts are close to each other on the whole dataset). We are open to adding more statistics following any suggestions from the reviewers.
> 4. We added the remark in the paper that we used mutually non-exclusive categories with videos from the dataset (User-Generated Content, Shaking, Sports, Nature, Gaming / Animation, Low Bitrate (up to 1,000 kbps) and High Bitrate (above 6,000 kbps)). This means that UGC videos could be classified as e.g. nature or sports. We assigned categories during an in-lab assessment by 5 people.

---

### Official Review · Reviewer_q4yp · 2022-07-23
**A valuable dataset and benchmark for video quality metrics**

**Rating:** 7
**Confidence:** 3
**Clarity:** The paper is written well.

**Strengths:**

The authors carefully select videos, distortions, types of new encoding standards, collect subjective scores, and evaluate a wide range of quality metrics.

**Weaknesses:**

The authors argue that we need new benchmarks because existing benchmarks are commonly investigated using older compression standards, such as AVC. It implies that the artifacts generated by newer compressors may be differently distributed from the older ones and such distribution shift will affect the performance of video quality metrics. However, this claim is not evidenced in the motivation part or echoed in the results section. The results in Table 3 and Table 4 show that exiting metrics (e.g., VMAF) perform good even with the newer Codec with above or around 0.9 SROC. This weakens the necessity of constructing a new benchmark with newer Codecs.

**Additional Feedback:**

NA.

**Correctness:**

The dataset is constructed in a sound way. Since subjective scores are collected via a crowdsource platform, I am little concerned about whether the viewers' environments (e.g., devices, screen size, network quality, etc.) affect the scores. It would be better to add more details on how the viewers give their scores.

**Documentation:**

The benchmark and dataset are well structured and documented.
On question about the dataset, will the open part of the dataset be publicly accessed (if not yet)?

**Ethics:**

No ethics concerns from my side.


**Relation To Prior Work:**

the related works are well discussed.

**Summary And Contributions:**

The authors construct a new benchmark of video quality metrics on a wide range of video compression methods. It considers more video encoding standards and collects subjective scores via crowd sourcing.

---

> ### Author Response · Authors · 2022-08-17
> **Response to Reviewer q4yp**
>
> Thank you for your review and suggestions! We have the following comments:
> 1. On the necessity of a new benchmark while existing metrics (e.g., VMAF) perform good even with the newer codec.
> Many of newly developed metrics claim state-of-the-art results, while only a few of them can be reproduced on new datasets. For example, in the original paper, ST-GREED showed better results than VMAF https://arxiv.org/pdf/2010.13715.pdf . We aim to verify the current metrics for the compression task and to show the leaders in different categories, including new coding standards.
> Also, the leaders for different standards are not always the same, e.g. for new VVC coding, LINEARITY beated MDTVSFA and SSIM was better than VMAF. We also added an example (Figure 1: Crops from video sequences encoded by old and new encoding standards) that illustrates that new encoding standards provide new types of artifacts. We believe that these differences and new trends such as NN usage in video compression require additional tracking of metrics performance on it.
> 2. We included the details on subjective comparisons in Supplementary materials. There is a scheme of subjective experiments, as well as screen size, age and country statistics on pages 17, 18, 19 (2.3 Collection process).
> 3. The open part is already available by a secure link, it is included in the paper and information about our submission on openreview. The link will also be published on a benchmark page.

---

### Official Review · Reviewer_oqgs · 2022-07-28
**A very useful dataset and benchmark for video and image quality.**

**Rating:** 7
**Confidence:** 2
**Clarity:** The paper is well-written.

**Strengths:**

1. A new dataset and benchmark for video compression, produced by new compression standards.
2. Extensive answers to the dataset

**Weaknesses:**

1. The scale is relatively small (as shown table 1)
2. It's unclear how the video is selected (based on what methodology)


**Additional Feedback:**

Please refer to the section above.

**Correctness:**

The claims are correct and the dataset is constructed in a sound way (but lack details on how they are collected from vimeo)

**Documentation:**

The dataset is hosted on a website and it's pretty well maintained.

**Ethics:**

No ethical concerns were identified.

**Relation To Prior Work:**

The related works are described in section 2 and the paper clearly discusses how it differs from previous work.

**Summary And Contributions:**

The paper introduces a dataset and benchmark for video and image quality of video encoders. The dataset consists of compressed videos and subjective scores, collected via crowd-sourced comparison. The benchmark consists of 2045 videos and 10,800 viewers and the authors provide assessment results on the open parts of the dataset.

---

> ### Author Response · Authors · 2022-08-17
> **Response to Reviewer oqgs**
>
> Thank you for your valuable feedback!
>
>  We added more information on video collection process into the paper and its supplementary materials (section “3.2 Video dataset” and “2.3 Collection process” in supplementary).
>
> We agree that the scale of our dataset is not the largest in terms of the amount of videos. We aimed to represent compression-related distortions and used not only several encoders to provide different artifacts, but also different encoding presets and bitrates. Due to the significant expenses required for subjective comparisons, we haven’t beaten the existing dataset by the number of original videos yet, but we will enlarge the video collection gradually, and add more videos and codecs. Besides, in our method we don’t measure correlations with subjective scores on all dataset, but measure correlation independently for compressed streams of one reference video, and then average the correlation via Fisher z-transformation. Even with the current amount of original videos we received small confidence intervals.

---

### Official Review · Reviewer_Wvxs · 2022-07-28
**Video Quality Metrics Benchmark**

**Rating:** 6
**Confidence:** 4
**Correctness:** Yes
**Clarity:** Yes

**Strengths:**

The paper is well written and provides clarity of thought. It provides new avenues by introducing data for other encoding standards as well. The subjective scores have been computed on the basis of a large pool of responses. The evaluation scheme used is thorough.

**Weaknesses:**

The results section could have been written in a more lucid and better way. The tables in the paper could have been explained a bit more. Information about the process of video collection is missing, i.e. queries used for video collection, genres of videos, etc. The authors have not briefly explained the application field and the larger impact of the proposed work.

**Additional Feedback:**

..

**Documentation:**

Yes, sufficient and well structured documentation has been provided.

**Ethics:**

No, there are no ethical concerns for this dataset

**Relation To Prior Work:**

Yes

**Summary And Contributions:**

The paper introduces a new dataset of video quality metrics for many video compressions. The dataset consists of various videos encoded by a wide variety of standards. The authors also provide subjective scores and results for various evaluated metrics.

---

> ### Author Response · Authors · 2022-08-17
> **Response to Reviewer Wvxs**
>
> Thank you for your review!
> Following your suggestions, we have updated the following sections: “3.2 Video dataset”, “3.5 Results”, and “4 Conclusion”.
> * In “3.2 Video dataset” we added details about the process of video collection: we searched for the widest possible number of videos from Vimeo using search keywords such as "a", "the", "of", "in", "be", "to", etc. This allowed us to download videos of a variety of genres (sports, gaming, nature, interviews, UGC, etc.). Supplementary materials provide links to source videos and more information about video collection (2.3 Collection process).
> * In “3.5 Results” we organized conclusions by the results illustrates in different tables.
> * In “4 Conclusion” we illustrated possible application fields and described possible impact of our contribution.
>   - The dataset will be useful for developers, researchers and industry experts when they evaluate new metrics performance for video compression quality estimation
>   - The dataset can be used for training or fine-tuning of new quality assessment models to achieve more precise results for video compression quality assessment.
>   - Our benchmark will remain as an unbiased test of new image- and video-quality metrics for video compression quality assessment.

---

### Meta-Review · Area_Chair_WswJ · 2022-09-09

**Recommendation:** Accept
**Confidence:** 4

**Metareview:**

The submission receives 5 reviews form 5 reviewers and all are positive about the proposed dataset presented in the submission. AC reads all reviews and discussions and agree with the reviewers. AC recommends to accept the submission as a spotlight.

---

### Decision · Program_Chairs · 2022-09-16

Accept